# A diversity-oriented rhodamine library for wide-spectrum bactericidal agents with low inducible resistance against resistant pathogens

Xiao Luo[1,2], Liujia Qian[3,4], Yansheng Xiao[2], Yao Tang[2], Yang Zhao[5], Xia Wang[5], Luyan Gu[2], Zuhai Lei[2], Jianming Bao[4], Jiahui Wu[4], Tingting He[2], Fupin Hu[6], Jing Zheng[5], Honglin Li[5], Weiping Zhu[1,2], Lei Shao[4], Xiaojing Dong[4], Daijie Chen[3,4], Xuhong Qian[1,2] & Youjun Yang [1,2]

Antimicrobial resistance is a public health emergency and warrants coordinated global efforts. Challenge is that no alternative molecular platform has been identified for discovery of abundant antimicrobial hit compounds. Xanthene libraries have been screened for bioactive compounds. However, the potentially accessible chemistry space of xanthene dyes is limited by the existing xanthene synthesis. Herein we report a mild one-step synthesis, which permits late-stage introduction of a xanthene moiety onto i.e. natural products, pharmaceuticals, and bioactive compounds and construction of a focused library of rhodamine dyes exhibiting facile functional, topographical and stereochemical diversity. In vitro screening yields 37 analogs with mid-to-high bactericidal activity against WHO priority drug-resistant pathogens. These findings suggest that synthetic dye libraries exhibiting high structural diversity is a feasible chemical space combating antibacterial resistance, to complement the natural sources.

[1] State Key Laboratory of Bioreactor Engineering, East China University of Science and Technology, Shanghai 200237, China. [2] Shanghai Key Laboratory of Chemical Biology, School of Pharmacy, East China University of Science and Technology, Shanghai 200237, China. [3] School of Pharmacy, Shanghai Jiao Tong University, Shanghai 200240, China. [4] State Key Laboratory of New Drug and Pharmaceutical Process, Shanghai Institute of Pharmaceutical Industry, Shanghai 201203, China. [5] Shanghai Key Laboratory of New Drug Design, School of Pharmacy, East China University of Science and Technology, Shanghai 200237, China. [6] Institute of Antibiotics, Huashan Hospital, Fudan University, Shanghai 200040, China. These authors contributed equally: X.L., L.J.Q. Correspondence and requests for materials should be addressed to Y.Y. (email: youjunyang@ecust.edu.cn) or to D.C. (email: hccb001@163.com) or to X.Q. (email: xhqian@ecust.edu.cn)

Antibiotics have revolutionized medical practices, have dramatically increased the average life expectancy of humans and are among the greatest medical break-throughs of the last century. However, antibiotics have only a limited period of clinical-utility before offset by the inevitable emergence of resistance[1]. For example, it took only a few years before resistance toward many antibiotics was discovered, such as penicillin, gentamicin, and ceftazidime[2]. Vancomycin lasted longer before resistance emerged, partly because it was typically used as the last resort. Nevertheless, vancomycin intermediate/resistant pathogens are now widespread[3]. New antibiotics are desperately sought after to fight bacterial resistance[4]. Ironically, large pharmaceutical companies have dramatically shrunk or cut their antibiotic programs. The reasons for this situation are manifold. The golden mine for antibiotics, the secondary meta-bolites of soil actinomycetes, has run dry after decades of exploitation[5,6]. An alternative chemical space for abundant antibiotic hit compounds is yet to be identified[7]. Third, antibiotic development has a low expected profit return. As a result, the antibiotic pipeline has been running at an alarmingly slow pace in the past few decades[8]. In particular, among these recently approved few antibiotics, most are actually chemically modified derivatives of the existing classes of drugs, most of which are of natural origin[9–12]. The resistant strains may readily mutate to resist these analogs if their existing resistance mechanisms do not already exhibit partial cross-effectiveness[13–16]. The genes encoding resistance to the natural products that are present within the original organisms can also be horizontally transferred to pathogenic microbes enabling resistance to emerge. To prevent the post-antibiotic era becoming reality, alternative chemical space capable of producing abundant antibiotic scaffolds are warranted.

Previously untapped natural sources, i.e. uncultured soil microbes[17], ocean bacteria[18], unexpressed metabolites coded by silent operons[19], and human commensals[20], have recently been explored and yielded antibiotic compounds. At the same time, synthetic libraries should not be overlooked for antibiotic discovery[4]. Salvarsan for syphilis and prontosil for streptococcus were two classic antimicrobial drugs discovered from the synthetic dye libraries in the early 20th century, and other notable examples are quinolones and the recent addition of oxazolidinones[21]. Despite the fact that target-based antibiotic discovery since the 1990s has been largely unsuccessful[7,22], synthetic library of unique chemical space exhibiting high structural complexity and diversity offers opportunities for formulating new paradigms for antibiotic development[23–25]. Recently, the diversity-oriented fluorescence library approach (DOFLA) has been proven to be a robust method for discovery of bioactive molecules via pheno-typic screening[26]. Xanthene dyes, classic small-molecule fluor-ophores including fluorescein, rhodamine and rhodol, has recently attracted attentions for DOFLA. For example, Chang et al. and other research groups constructed a number of focused libraries based on various fluorescent scaffolds[27,28], and further discovered useful imaging agents or fluorescent probes for small-molecule targets, nucleic acids, proteins, lipid droplets, stem-cells and live animals[29–31]. Burgess et al. constructed a library of mitochondria-targeting rhodamine analogues exhibiting potent anti-proliferative effects toward tumor cells[32,33]. High structural diversity of dye libraries, the key to its success in the fluorescence library approach, is however the biggest obstacle to overcome. The ubiquitous structure feature of a dye is a two-dimensional conjugated system, to which some simple chemical groups may be attached. Therefore, dyes are typically not endowed with complex three-dimensional features and not ideal for discovery of bioactive compounds via phenotypic screening. Also, the conditions for dye synthesis are usually harsh and installation of

chemical groups exhibiting exotic functional, topological and stereochemical features are difficult[34–36]. Diversity-oriented rho-damine library could not be constructed with the conventional method due to, extensive side-reactions, tedious purification, low yields and most important of all limited scope of compatible substrates. Take the xanthene dye library by Ahn et al.[37] as an example. It is a chemical library of 300+ compounds. However, all follow the general feature, i.e. a xanthene core with a small aromatic group at its C-9. Therefore, the library is not high in terms of diversity. A diversity index of only 3.228 was calculated for this library of rhodamine dyes with Extended-Connectivity FingerPrint (ECFP)[38–40], which is a convenient estimate of the structural diversity of a chemical library (vide infra).

We herein report the construction of a diversity-oriented rhodamine library and discovery of potent and wide-spectrum bactericidal agents without inducible resistance. Phenotypic screening of the library yields two promising hit compounds, and **RD22** and **RD53** were found to be particularly noteworthy through further studies. They inhibit the growth of a wide-spectrum of pathogens, with a potent MIC of 0.5–2 μg mL$^{-1}$ against the Gram-positive strains, and a MIC of 2–16 μg mL$^{-1}$ against the Gram-negative strains. They are potent bactericidal agents and pathogens do not readily acquire resistance by geno-mic mutations toward **RD22** and **RD53**. They exhibit low hae-molytic activity with a Lysis20 over 100 μg mL$^{-1}$. The aforementioned properties of **RD22** and **RD53**, along with other less potent analogs suggest that the rhodamine library is a viable source of antibacterial hit compounds.

## Results

**Construction of a library with 70 rhodamine analogs.** A number of preparative methods are available for xanthene dyes. The earliest synthesis was by von Baeyer[34] in 1871 through the condensation of phthalic anhydride with phenol at high tem-perature, with Lewis acid or strong proton acid as the catalyst. To accommodate the growing demands for functional xanthene dyes, an alternative mild preparation was introduced by Tsien et al. in 1989, involving nucleophilic addition of a phenyl Grignard (or lithium) reagent with a xanthenone at low temperature[36]. This has enabled the preparation of the library of Ahn et al.[37]. Though robust, diversity of the xanthene dyes from this method is still limited by the availability and structural versatility of Grignard reagents. The limitation remains unsolved with the report of two-step Grignard/demethylation method by Yang et al.[41]. To address those difficulties with the classic xanthene synthesis, we have developed a mild one-step and high-yielding xanthene synthesis (Fig. 1a) via nucleophilic condensation of a dilithium reagent (**2**) with readily available esters, anhydrides, or even some amides (Supplementary Figure 1). This method is versatile and allows the preparation of xanthene dyes with exotic substituents at C-9, e.g. phenyls, aryls, alkyls, alkenyls and acyls (vide infra) (Fig. 1). Shortly after we filed a Chinese invention patent[42] of this method in March 2017, Grimm et al.[43] and Fischer and Sparr[44] also independently developed and reported this method in recent literature.

The capability of this method in preparation of rhodamines with substituted phenyl group at C-9 was first examined (Fig. 1b). As a proof-of-concept experiment, the dilithium reagent (**1**) was reacted with an alkyl benzoate, i.e. methyl benzoate (**S1**), to furnish **RD1** in an 83% yield. Rhodamines with a methyl (**RD2**), hydroxylmethyl (**RD3**), formal (**RD5**) or a carboxyl group (**RD6**) installed at the *ortho*-position of the bottom phenyl ring was also smoothly prepared in excellent yields by reacting **1** with 2-methyl-benzoate (**S2**), phthalide (**S3**), 3-methoxyphthalide (**S4**) or phthalic anhydride (**S6**). **RD4**, rather than **RD5**, could be

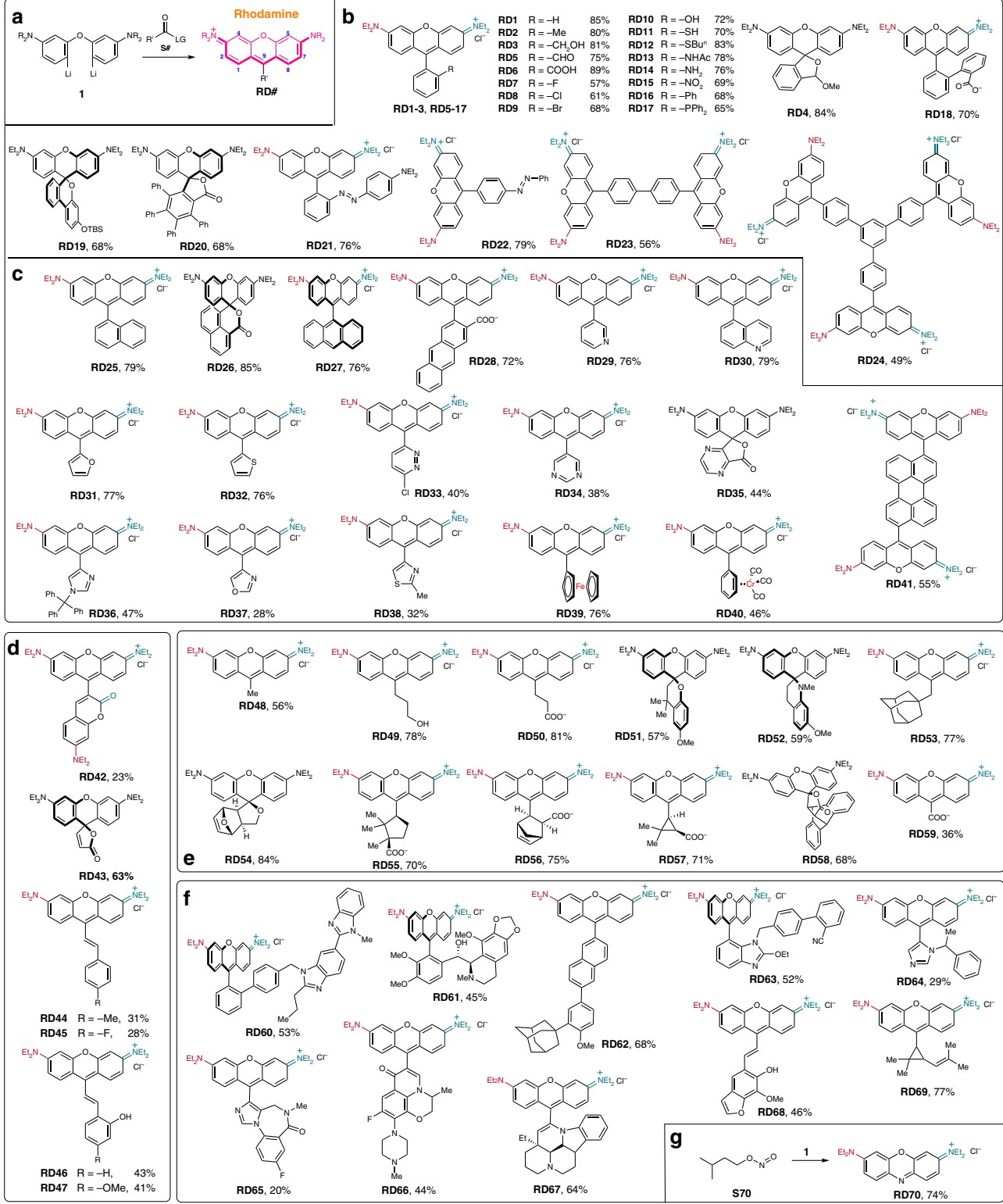

**Fig. 1** The rhodamine library via nucleophilic condensation of a dilithium reagent **1** with various substrates. **a** The general synthetic pathway. Structures of rhodamine dyes with **b** phenyl groups at C-9 (**RD1**-**RD24**); **c** polyaromatic or heteroaryl groups at C-9 (**RD25**-**RD41**); **d** alkenyl groups at C-9 (**RD42**-**RD47**); **e** alkyl groups at C-9 (**RD48**-**RD59**); **f** bioactive molecules at C-9 (**RD60**-**RD69**) and **g** oxazine (**RD70**). Most of the rhodamine dyes are in their chloride salt form

obtained in an 84% yield if the reaction was quenched with saturated $NH_4Cl$, rather than dilute HCl solution.

The one-step high-yielding synthesis of **RD3** and **RD5** shown here is superior to their literature preparations[45,46]. The *ortho*

fluoro-, chloro- or bromo-substituted methyl benzoate (**S7-S9**) were compatible with this method to give **RD7-RD9** in good to excellent yields. However the attempt with methyl 2-iodobenzoate was complicated with deiodonation leading to the

formation of **RD1**, presumably via metal-halogen exchange. Rhodamines with hydroxyl, sulfhydryl, acetyl, amino and nitro groups (**RD10**, **RD11, RD13, RD14** or **RD15**) were readily prepared by reacting **1** with benzo[*d*][1,3]dioxin-4-one (**S10**), benzo[*d*][1,3]oxathiin-4-one (**S11**) or benzo[*d*][1,3]oxazin-4-one (**S13**), methyl-2-nitrobenzoate (**S15**) respectively. Synthesis of rhodamine dyes with bulky groups at the *ortho* position of the bottom phenyl ring is not trivial. This method was powerful in this regard. Reactions between the dilithium reagent **1** and **S16**, **S17, S18, S19** and **S20** (Supplementary Figure 1) have produced a series of rhodamine dyes with such bulky groups as phenyls in **RD16**, **RD18**, **RD19** and **RD20**, and diphenylphosphoryl in **RD17**, in a yield ranging from 65 to 70%. The carbonyl of **S18** is involved in electronic push-pull system and therefore is not very reactive toward nucleophilic attack. Still, **S18** was a viable substrate to react with **1** furnishing a rhodamine (**RD18**) with an electron-rich phenyl ring at the *ortho*-position, in a 70% yield. **RD20** is a rhodamine modified with four freely rotatable phenyl groups and structurally analogous to an AIE-luminogen[47], e.g. tetraphenylethene. Its potentials in AIE-based studies warrant further investigations. Rhodamines with phenyl diazo groups (**RD21** and **RD22**) were also successfully prepared in yields over 76%. Unusual structures like **RD23** or **RD24**, expected to exhibit high molar absorptivity and fluorescence brightness were successfully obtained, in good yields of 56% and 49%, respectively.

Onto the C-9 position of the rhodamine core can be installed aryl groups other than phenyl, e.g. polyaromatic moities, heterocyclic aromatic moieties and the exotic sandwich/half-sandwich complexes (Fig. 1c). Rhodamines with naphthalene (**RD25**, **RD26)** or anthracene (**RD27**, **RD28)** were prepared from reactions of **1** with the corresponding ester or anhydride in good isolated yields (typically over 75%). Esters of pyridine, quinoline, furan, thiophene, pyridazine, pyrimidine, pyrazine, imidazole, oxazole, and thiazole (Supplementary Figure 1) could also react with the dilithium reagent **1** to prepare the corresponding rhodamine dyes (**RD29-RD38**). With the methyl ester of ferrocene, we prepared **RD39**, in which the ferrocene moiety is directly bonded to the C-9 position of the rhodamine core. We further showed that the half-sandwich complex could be directly attached to the rhodamine core (**RD40**). The potential applications of **RD40** in sensing, imaging and catalysis are attractive. Substrates bearing two ester moieties could be used to template multiple rhodamine cores, to give **RD41**.

Alkenyl groups can also be installed onto the C-9 position of a rhodamine core (Fig. 1d). The double bond in such rhodamine dyes exhibits facile chemistry and may be useful in site-specific labeling[48], chemosensing of nucleophiles[49] and reactive oxygen species[50] and optical switching[51]. With the ester of a coumarin, a rhodamine-coumarin dye pair (**RD42**) was synthesized as one example of this group of rhodamine dyes. Reactions between the dilithium reagent **1** and a range of substrates like maleic anhydride, cinnamates, or coumarins (Supplementary Figure 1) were all capable of furnishing such alkenyl substituted rhodamine dyes, i.e. **RD43-RD47**. We note that the double bond of **RD43** adopts *cis*-conformation, as suggested by a coupling constant of 5.4 Hz between the two alkenyl H's. The double bonds in **RD44-47** are all in *trans*-conformation, regardless the *cis*- or *trans*-conformation of the double bonds in the corresponding starting material (**S44-S47**, Supplementary Figure 1).

Alkyl esters, lactones and anhydrides are all feasible substrates for synthesizing rhodamine dyes and diverse range of complex alkyl groups are also conveniently installed to the C-9 of the rhodamine core (**RD48-RD59**) (Fig. 1e).

Till this point, we have demonstrated that this rhodamine synthesis is convenient, powerful and reliable. The range of substrates compatible for this method is exceedingly broad, including esters, anhydride, lactone, aryl lactam, and potentially more. To further promote the scope of this rhodamine synthesis, we are particularly interested in the possibility of using ester-/ lactone-/anhydride-containing bioactive molecules, e.g. pharmaceuticals and pesticides (**S60-S69**, Supplementary Figure 1), in rhodamine synthesis. The structures of such compounds are typically complex and there were no attempts to use them in dye synthesis. Though probably unexpected to the field, many of such compounds, e.g. telmisartan methyl ester (**S60**), noscapine (**S61**), adapalene (**S62**), a precursor of candesartan (**S63**), etomidate (**S64**), the methyl ester of flumazenil (**S65**), levofloxacin methyl ester (**S66**), vinpocetine (**S67**), methoxsalen (**S68**), ethyl chrysanthemate (**S69**) were indeed viable substrates for this rhodamine synthesis (Fig. 1f).

This method is also suitable for preparation of an oxazine dye **RD70** (Fig. 1g) from isoamyl nitrite, which differs from its rhodamine analog in that it bears a nitrogen atom rather than a methine carbon.

**Diversity index**. The extended-connectivity fingerprint (ECFP) method was employed to assay the diversity of our library because of its wide-acceptance in drug discovery and has been integrated into various commercial softwares. ECFPs encode each atom and its molecular environment within a circle with a diameter of varying radius of chemical bonds. Depending on the chosen bond radius, different numbers of structural features having different sizes are produced. In this study, we introduce the definition of the total number of fingerprint features as a quantitative measure of the structural diversity of a library. We calculated the specified fingerprint for each molecule using ECFP_6 and counted the total number of unique fingerprint features collected over all the molecules. Then the Diversity Number Finger Print Features (DNFPF value) was defined as the total number of fingerprint features divided by the number of molecules. A DNFPF value of 18.875 was calculated for the obtained library of rhodamine dyes (**RD1-RD69**), while a DNFPF value of 3.228 was found for the Ahn's library[37] of rhodamine dyes (Supplementary Table 6). This verified that the synthetic method presented herein is feasible for construction of a rhodamine library with much enhanced structural diversity.

**Fluorescence properties**. The fluorescence properties of **RD1-RD70** were studied predominantly in PBS (50 mM, pH = 7.4 containing 1% DMSO) with UV-Vis absorption and fluorescence spectroscopies (Fig. 2 and Supplementary Table 1). Other solvents were not tested because we are typically concerned with the potentials of this library in bio-oriented applications. The maximal absorption wavelengths of rhodamine dyes with alkyl groups (**RD48-RD59**) at C-9 typically absorb at shorter wavelength region of 545–560 nm and ones substituted with heteroaryl groups (**RD25-RD41**) absorb at longer wavelength region of 551–585 nm, while others absorb in the range of 560–570 nm. Most dyes exhibit a small Stokes shift of *ca.* 15–30 nm, which is typical of a rhodamine, some have displayed a Stokes shift larger than 30 nm, e.g. 34 nm for **RD31**, 41 nm for **RD33**, 37 nm for **RD35**, and 34 nm for **RD41**, 32 nm for **RD47**, 39 nm for **RD53**, 35 nm for **RD56**, 42 nm for **RD59**. The fluorescence brightness of many dyes of this library is comparable or higher than that of the rhodamine B (**RD6**). They are useful for cell imaging-based applications (Supplementary Figure 72–131). Many others are much less fluorescent. Dim dyes may be used as fluorescent probes, in case their fluorescence may be chemospecifically turned on. For example, we have found that **RD17** can respond to nitroxyl (HNO) and induce a *ca.* 5 fold fluorescence turn-on

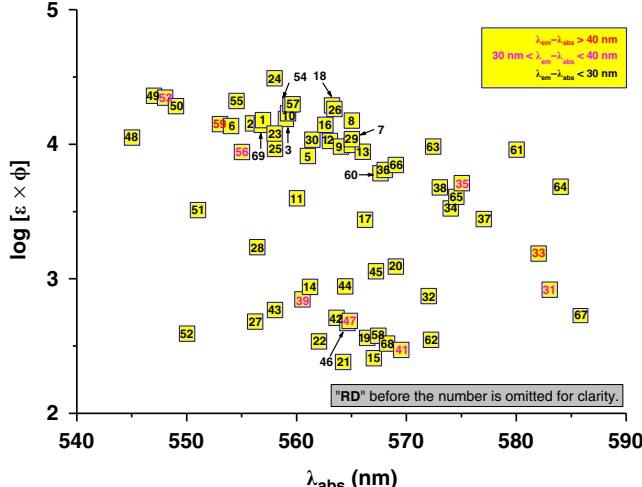

**Fig. 2** Fluorescence properties of the rhodamine library in PBS buffer. The y-axis is the brightness, which is expressed as log ($ε × Φ$) of a given rhodamine dye and the x-axis is the absorption maximum. **RD23** and **RD24** were tested in EtOH

(Supplementary Figure 70) and **RD28** may scavenge $^1O_2$ and produce a fluorescence turn-on of *ca.* 14 fold (Supplementary Figure 71).

**Antibacterial studies of the rhodamine library**. The antimicrobial activity of (**RD1-RD70**) were preliminarily evaluated against two drug-resistant bacteria, which were a strain of Gram-positive methicillin-resistant *Staphylococcus aureus* (MRSA, ATCC43300) and a strain of Gram-negative *Acinetobacter baumannii* (ATCC19606). The minimum inhibitory concentrations (MIC) of each compound were measured using the standard CLSI broth protocol (Fig. 3, Supplementary Table 2). Typically, antimicrobial potency of a compound is considered mid-to-high if its MIC is 8 μg mL$^{-1}$ or lower. Out of the total 70 analogs, 37 were found to exhibit a MIC of 8 μg mL$^{-1}$ or lower against MRSA, and 5 aganist *A. baumannii*. This is a result much to our delight as well as to our surprise considering the relentless yet largely futile efforts that the community has dedicated to the quest for a molecular platform for discovery of abundant compounds with antimicrobial activity. Further analysis revealed that these bioactive analogs are quite scattered in terms of the nature of substituent at the C-9 position of the rhodamine core. Among these bioactive analogs, 14 analogs bear a phenyl group at C-9 position, 12 a heteroaromatic moiety, 6 an alkenyl and 5 an alkyl group. Presumably, the rhodamine core is the pharmacophore and the substituent at the C-9 playing a subsidiary, yet critical role. These results strongly suggested that this diverse-oriented rhodamine library is a viable platform for discovery of potent antimicrobial agents.

Though all these 37 + 5 compounds with mid-to-high antibacterial activity worth further pursuing, we prioritized the ones with a higher activity for in-depth study of their antibiotic spectrum (Table 1). Those compounds with the lowest MIC's against ATCC43300 (**RD1/15/22/31/45/46/53**) or ATCC19606 (**RD12/22/44/45/53**) were chosen for further screening against three more Gram-positive bacteria, including the ATCC25923 (methicillin-sensitive *Staphylococcus aureus*, MSSA), ATCC51299 (vancomycin-resistant *Enterococcus faecalis*, VRE), and ATCC29212 (vancomycin-sensitive *Enterococcus faecalis*, VSE), and three Gram-negative bacteria, including ATCC13883 (*Klebsiella pneumoniae*), ATCC25922 (*Escherichia coli*) and ATCC27853 (*Pseudomonas aeruginosa*) (Table 1). **RD22/46/53**

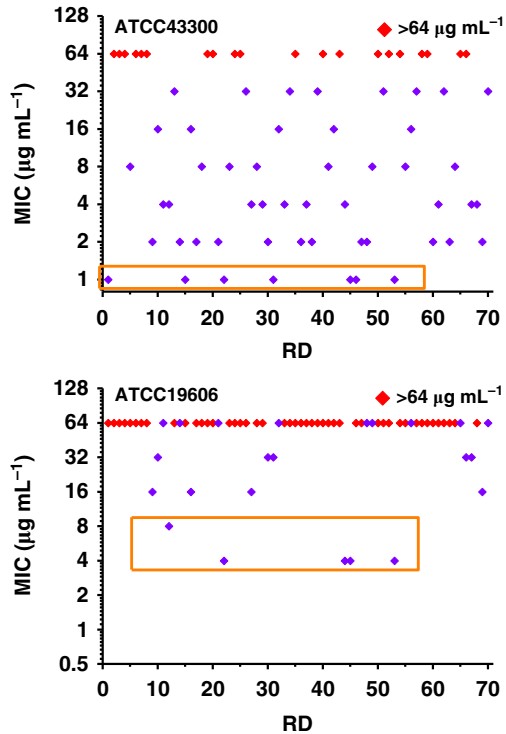

**Fig. 3** Activity of **RD1**-**RD70** against *MRSA* and *Acinetobacter baumannii*. The MIC (μg mL$^{-1}$) of each compound was measured by the standard CLSI broth protocol

showed potential broad-spectrum Gram-positive antibiotic activity with a low MIC of 0.5–2 μg mL$^{-1}$, while **RD1/15/31/45** were much less active toward *E. faecalis* compared to their activity against *S. aureus*. **RD22/44/53**, but not **RD12/45**, also exhibited antibiotic activity toward Gram-negative bacteria other than ATCC19606, with a relatively higher MIC of 8–32 μg mL$^{-1}$. From this study, **RD22/53** stood out as the two most attractive hits. First, their MIC's against MRSA is 1 μg mL$^{-1}$, the lowest among this shortlist, and comparable or even lower than those of vancomycin and linezolid (Table 1), the last resorts for MRSA infections. Second, they were also equally potent toward both vancomycin-sensitive and vancomycin–resistant *E. faecalis* with a MIC of 2 μg mL$^{-1}$. Third, they were also active toward various clinically significant Gram-negative bacteria with MIC of 4–16 μg mL$^{-1}$. These encouraging preliminary results promoted us to explore the antibiotic profile of **RD22/53** in a greater depth.

The antibiotic spectrum of **RD22** and **RD53** were further checked against a panel of standard reference and clinical-isolated Gram-positive and Gram-negative strains, which encompass most of the 12 deadliest drug-resistant bacteria[52] classified by the World Health Organisation (WHO) (Table 2). The Gram-positive strains include *S. aureus* (MRSA, MSSA), *E. faecalis* and *E. faecium* (VRE and VSE), *S. epidermidis*, *S. pyogenes*, *H. pylori*, *S. pneumonia*, while the tested Gram-negative strains are *A. baumannii*, *P. aeruginosa*, *E. coli.*, *K. pneumonia*, and *S. flexneri*. **RD53** exhibited a potent MIC of 0.25–2 μg mL$^{-1}$ toward all the aforementioned Gram-positive strains, regardless of the drug-resistance profile of these bacteria. The potency of **RD53** toward the Gram-negative strains was typically lower compared to the Gram-positive strains. For example, a MIC of 4 μg mL$^{-1}$ was observed for *A. baumannii*, and 16 μg mL$^{-1}$ for *P. aeruginosa*, 2–8 μg mL$^{-1}$ for various strains of *E. coli*, 4–16 μg mL$^{-1}$ for *K. Pneumoniae* and 1 μg mL$^{-1}$ for *Shigella flexneri*. **RD53** was not only found to be effective toward standard reference strains, but

**Table 1 Antimicrobial activity of selected RD dyes against four Gram-positive and four Gram-negative bacteria**

| MIC (µg mL$^{-1}$) | Gram-positive organism | | | | Gram-negative organism | | | |
|---|---|---|---|---|---|---|---|---|
| | ATCC43300 | ATCC25923 | ATCC51299 | ATCC29212 | ATCC19606 | ATCC13883 | ATCC25922 | ATCC27853 |
| RD1 | 1 | 0.5 | 16 | 16 | / | / | / | / |
| RD12 | / | / | / | / | 8 | 128 | 64 | 128 |
| RD15 | 1 | 0.5 | 32 | 16 | / | / | / | / |
| RD22 | 1 | 0.5 | 2 | 2 | 4 | 16 | 16 | 16 |
| RD31 | 0.5 | 0.5 | 16 | 16 | / | / | / | / |
| RD44 | / | / | / | / | 4 | 32 | 8 | 16 |
| RD45 | 1 | 1 | 8 | 4 | 4 | 64 | 16 | 16 |
| RD46 | 1 | 1 | 2 | 2 | / | / | / | / |
| RD53 | 1 | 1 | 2 | 2 | 4 | 16 | 8 | 16 |
| vancomycin | 2 | 2 | / | 4 | / | / | / | / |
| linezolid | 1 | / | 2 | 2 | / | / | / | / |
| polymyxin E | / | / | / | / | 1 | / | / | 1 |
| tigecycline | 2 | / | 1 | / | 2 | / | / | / |
| daptomycin | 2 | / | 4 | / | / | / | / | / |

**Table 2 Antibacterial activity of RD22 and RD53 against a panel of Gram-positive and Gram-negative pathogens**

| | Organism | | MIC$^a$ | |
|---|---|---|---|---|
| | | | RD22 | RD53 |
| Gram+ | S. aureus | ATCC25923(MSSA$^b$) | 1 | 1 |
| | | CMCC26003(MSSA$^b$) | 1 | 0.5 |
| | | ATCC43300 (MRSA$^b$) | 1 | 1 |
| | | 10 Clinical isolates(MRSA$^b$) | 0.25 | 0.25–2 |
| | E. faecalis | ATCC29212(VSE$^d$) | 2 | 2 |
| | | ATCC51299(VRE$^d$) | 2 | 2 |
| | | 5 Clinical isolates(VSE$^d$) | 1 | 1 |
| | E. faecium | ATCC35667 | 1 | 2 |
| | | 5 Clinical isolates(VRE$^d$) | 1 | 1 |
| | | 3 Clinical isolates(VSE$^d$) | 1 | 0.5–1 |
| | S. epidermidis | CMCC26069(MSSE$^c$) | 1 | 1 |
| | | 2 Clinical isolates(MSSE$^c$) | 1 | 1 |
| | | 1 Clinical isolates(MRSE$^c$) | 1 | 1 |
| | S. pyogenes | CMCC32006 | 1 | 1 |
| | | 3 Clinical isolates | 0.5 | 2–4 |
| | H. pylori | Sydney Strain 1 | n.a. | 0.5 |
| | S. pneumoniae | 3 Clinical isolates (PRSP)$^e$ | 0.5–1 | 2–4 |
| Gram- | A. baumannii | ATCC19606 | 4 | 4 |
| | P. aeruginosa | ATCC27853 | 16 | 16 |
| | E. coli | ATCC25922 | 16 | 8 |
| | | 9 Clinical isolates(ESBL+)$^f$ | 2–4 | 2–4 |
| | | 4 Clinical isolates(ESBL-)$^f$ | 4 | 4–8 |
| | K. pneumoniae | ATCC13883 | 16 | 16 |
| | | 3 Clinical isolates(ESBL+)$^f$ | 4 | 4–8 |
| | | 4 Clinical isolates(ESBL-)$^f$ | 4–8 | 4 |
| | | 4 Clinical isolates(CRE)$^g$ | 4–8 | 8–16 |
| | S. flexneri | ATCC51081 | 2 | 1 |

$^a$MIC's are given in unit of µg mL$^{-1}$
$^b$MSSA (or MRSA): methicillin-sensitive (or resistant) *Staphylococcus aureus*
$^c$MSSE (or MRSE): methicillin-sensitive (or resistant) *Staphylococcus epidermidis*
$^d$VSE (or VRE): vancomycin sensitive (or resistant) *Enterococcus*
$^e$PRSP: penicillin-resistant *Staphylococcus pneumonia*
$^f$ESBL: extended spectrum beta-lactamases
$^g$CRE: carbapenem-resistant *Enterobacteriaceae*

also effective toward a broad range of clinically isolated, difficult-to-treat, multidrug-resistant bacteria, including *S. aureus* (MRSA, 10 strains), *E. faecalis* (VSE, 5 strains), *E. faecium* (VRE, 5 strains; VSE, 3 strains), *S. epidermidis* (MSSE, 2 strains; MRSE, 1 strain), *S. pyogenes* (3 strains), *S. pneumonia* (3 strains), *E. coli* (ESBL+, 9 strains; ESBL-, 4 strains), *K. pneumonia* (ESBL+, 3 strains, ESBL-, 4 strains, and CRE, 4 strains). Compared to **RD53**, **RD22** displayed equally potent antimicrobial activity against all the aforementioned standard reference Gram-positive and Gram-negative type bacteria and clinical isolates as well.

The clinical significance of the first-line bacteriostatic antibiotics, e.g., vancomycin, linezolid, et al., warrants no further elaboration. However, in some serious clinical situations, such as endocarditis, meningitis, osteomyelitis, neutropenia, bactericidal action is necessary because of the reduced rates of metabolism and cell division, the poor immunologic competence or the poor drug penetration[53]. The bactericidal activity of **RD22** and **RD53** were compared to multiple first-line antibiotics using the time-kill assay (Fig. 4). Methicillin-resistant *S. aureus*, Vancomycin-resistant *E. faecalis*, and polymyxin E-resistant *A. baumannii* were chosen as the tested Gram-positive and Gram-negative strains, respectively. MRSA (ATCC43300) were grown to early exponential phase and challenged with 10× MIC of vancomycin, linezolid, tigecycline, respectively (Fig. 4a). In the first 4 h, bacteria concentration remained steady in the presence of vancomycin and linezolid and then gradually dropped by two-three orders of magnitude within 24 h. Tigecycline exhibited a higher degree of bactericidal activity than vancomycin and linezolid. In comparison, **RD22** (Fig. 4b) and **RD53** (Fig. 4c) were more potent bactericidal agents than vancomycin, linezolid or tigecycline. Bacteria challenged by as low as 2.5× MIC lowered the bacteria concentration by four order of magnitude of bacteria within as short as 2 h. **RD22** and **RD53** also exhibits bactericidal activities to VRE (ATCC51299) (Fig. 4e, f). They lowered the concentration of VRE by over 4 orders of magnitude within 2 h by (5–10)× MIC. In comparison, linezolid, tigecycline did not exhibit strong bactericidal activity and bacterial concentration dropped by less than 2 orders of magnitude within 24 h. Daptomycin was able to reduce the cell concentration by over 4 orders of magnitude within 8 h (Fig. 4d). Polymyxin E-resistant *A. baumannii* (MIC = 256 µg mL$^{-1}$, stored in the lab) was obtained by serial passages of cells at sub-MIC concentration of polymyxin E during resistance acquisition of pathogens. **RD53** at 10 × MIC was able to reduce the cell concentration by over 4 orders of magnitude within 6 h (Fig. 4i) while at the same dose of tigecycline and **RD22** reduced the cell concentration by around 3 orders of magnitude (Fig. 4g, h). And **RD53** at 8 × MIC could also reduce the cell concentration of *H. pylori* by more than 4 orders of magnitude within 2 h (Supplementary Figure 132). So, the time-kill assay proved that **RD22** and **RD53** are potent bactericidal agents. They can potentially complement bacterio-static antibiotics in serious bacterial infections to hosts, whose immune response is compromised.

**Resistance studies of RD22 and RD53**. Ease of resistance acquisition toward **RD22** and **RD53** was evaluated using MRSA, VRE and *A. baumannii* as tested bacteria (Fig. 5). Serial passages of MRSA in the presence of sub-MIC levels of levofloxacin over a

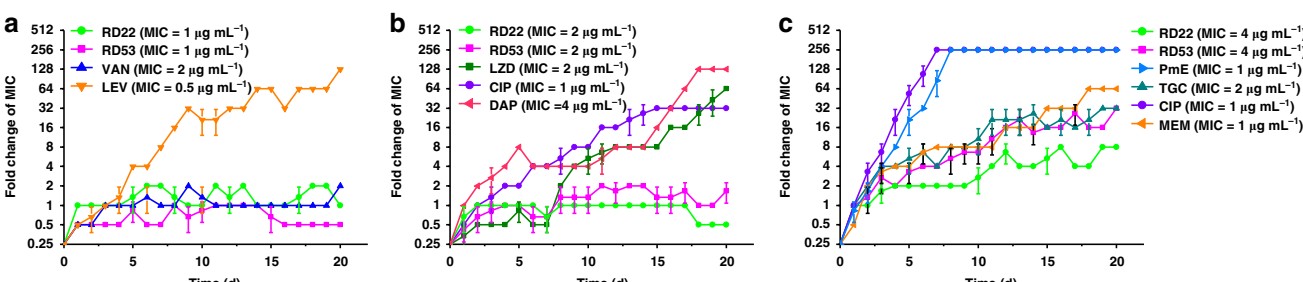

**Fig. 4** Time-dependent killing of pathogens by **RD22** or **RD53** compared to clinical antibiotics. MRSA (ATCC43300) challenged with **a** vancomycin, linezolid, tigecycline; **b RD22**; **c RD53**. VRE (ATCC51299) challenged with **d** vancomycin, linezolid, tigecycline, daptomycin; **e RD22**; **f RD53**. Polymyxin E-resistant *Acinetobacter baumannii*. challenged with **g** polymyxin E, tigecycline **h RD22**; **i RD53**. Pathogens were grown to early exponential phase and challenged with **RD22/53** or clinical antibiotics. Data are representative of three independent experiments. Error bars indicate s.d.

**Fig. 5** Resistance acquisition of pathogens during serial passaging in sub-MIC levels of antimicrobials. The y axis is the fold change of MIC. **a** MRSA (ATCC43300) towards **RD22** (MIC = 1 μg mL⁻¹, tested up to 2 × MIC), **RD53** (MIC = 1 μg mL⁻¹, tested up to 1 × MIC), levofloxacin (MIC = 0.5 μg mL⁻¹, tested up to 128 × MIC), and vancomycin (MIC = 2 μg mL⁻¹, tested up to 2 × MIC). **b** VRE (ATCC51299) toward **RD22** (MIC = 2 μg mL⁻¹, tested up to 1 × MIC), **RD53** (MIC = 2 μg mL⁻¹, tested up to 2 × MIC), linezolid (LZD, MIC = 2 μg mL⁻¹, tested up to 64 × MIC), ciprofloxacin (CIP, MIC = 1 μg mL⁻¹, tested up to 32 × MIC), and daptomycin (DAP, MIC = 4 μg mL⁻¹, tested up to 128 × MIC). **c** *A. baumannii* (ATCC19606) toward **RD22** (MIC = 4 μg mL⁻¹, tested up to 8 × MIC), **RD53** (MIC = 4 μg mL⁻¹, tested up to 32 × MIC), polymyxin E (PmE, MIC = 1 μg mL⁻¹, tested up to 256 × MIC), tigecycline (TGC, MIC = 2 μg mL⁻¹, tested up to 32 × MIC), and ciprofloxacin (CIP, MIC = 1 μg mL⁻¹, tested up to 256 × MIC), meropenem (MEM, MIC = 1 μg mL⁻¹, tested up to 64 × MIC). Data are representative of three independent experiments. Error bars indicate s.d.

period of 20 days raised the MIC by 128 folds, i.e. rendering the strain highly resistant to levofloxacin. In comparison, MRSA did not develop resistance toward vancomycin, exactly the reason why vancomycin serves as the last-line antibiotic against MRSA. It is very interesting to note that MRSA also did not acquire a resistance by genomic mutations toward **RD22** and **RD53** as well (Fig. 5a). In a separate experiment, VRE developed resistance toward linezolid, ciprofloxacin and daptomycin by serial passages in 20 days, with a raise of MIC's by 32–128 folds while no obvious resistance was observed with **RD22** and **RD53** by VRE (Fig. 5b). The Gram-negative *A. baumannii* readily acquired resistance by genomic mutations toward polymyxin E and ciprofloxacin with a 256 enhancement of MIC in as short as 7 days. Bacteria did develop resistance toward meropenem, tigecycline, **RD22** and **RD53** as well, but to a much reduced degree, i.e. 64-, 32-, 8-, and 32-fold enhancement of MIC within 20 days, respectively (Fig. 5c). It has become evident that either Gram-positive or Gram-negative pathogens were found not to readily acquire resistance by genomic mutations toward **RD22** and **RD53**. Analogously, *E. coli*, *E. faecalis* and *S. aureus* were found not to acquire resistance toward **RD53** (Supplementary Figure 133). Development of resistance toward an antibiotic by bacteria is an evolutionary inevitability and thereby imposes a limit on the lifespan of any given antibiotic. And resistance was involved in clinical failure of some antibiotic administration regimes. This is especially evident because the antibiotic levels at some points during treatment are in the sub-inhibitory range[54]. Therefore, the low tendency of resistance acquisition toward **RD22** and **RD53** by various pathogens greatly enhances their potentials for further drug development.

**Probing mechanism of action**. The mechanism of action was investigated, with **RD53** as an example. First, the lack of noticeable morphological variation of MRSA and *A. baumannii*, as indicated by the SEM images (Fig. 6) with and without **RD53**, suggested that **RD53** did not destroy either the cell wall or membrane. This observation was further corroborated by the fact that **RD53** exhibits low haemolytic activity with a Lysis20 value of above 100 μg mL$^{-1}$. Second, a resistant strain of *A. baumannii* to **RD53** (**RD53**-Ab-20th) was produced by serial passages of ATCC19606 exposed to sub-MIC concentration of **RD53** over a period of 20 days, with the MIC of **RD53** raised from 4 μg mL$^{-1}$ to 128 μg mL$^{-1}$. Yet, the phenotypic profile of **RD53**-Ab-20th with respect to susceptibility to other antibacterial drugs, e.g., tigecycline (TGC), ciprofloxacin (CIP), polymyxin E (PmE), levofloxacin (LEV), and ampicillin-sulbactam (SAM), remains unchanged compared to ATCC19606, suggesting the lack of cross-resistance between **RD53** and these antibiotics (Supplementary Table 9). Third, selective pressure on antibacterial targets could also be measured by genomic profile of mutational resistant bacteria. So, the genomic analysis was carried out for clues on the potential mechanism of action of **RD53**. Comparative analysis revealed the existence of eight nonsynonymous single nucleotide variants (SNV) and one nonframeshift deletion (Table 3), which belong to resistance/tolerance-associated genes encoding efflux pumps, metabolism, DNA replication or transcriptional regulation related proteins[55,56]. It is interesting to find the mutation in *rsmJ* which encodes 16S rRNA (guanine1516 - N2) – methyltransferase. Notably, the mutation in 16S rRNA methyltransferase has been shown to cause kasugamycin resistance in *E. coli*[57,58]. These clues together have formulated a solid framework for further investigation of the mechanism of action of **RD53**.

**Structure-activity relationship investigation**. The structure-activity relationship of these rhodamine-type antibiotic

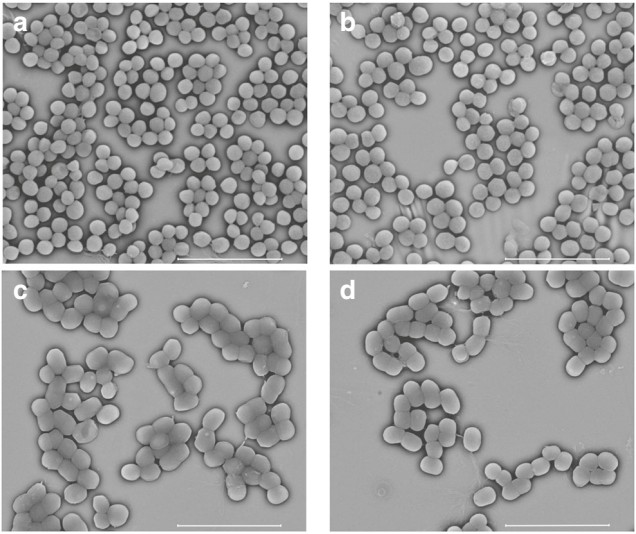

**Fig. 6** SEM images for MRSA and *A. baumannii* with or without treatment of **RD53**. **a** MRSA without treatment of **RD53**. **b** MRSA after incubation with 8 × MIC (8 μg mL$^{-1}$) **RD53** for 4 h. **c** *A. baumannii* without treatment of **RD53**. **d** *A. baumannii* after incubation with 8 × MIC (32 μg mL$^{-1}$) **RD53** for 4 h. SEM images were taken by FEI Nova NanoSEM 230. The SEM was operated in the high vacuum at an accelerating voltage of 7.5 kV. Scale bars represent 5 μm

compounds was examined. Out of the 70 members of this rhodamine library, 37 compounds exhibit antibiotic activity of varying potency (Fig. 3). This fact alone strongly suggested that their common structural motif is the pharmacologically active motif, which is the rhodamine core. To verify this hypothesis, we have prepared a few closely related analogs (**C1-2**, and **RD71-74**, Fig. 7). First, **C1** and **C2** differ from the **RD53** in that **C1** misses the oxygen atom of the rhodamine core compared to **RD53**, and **C2** has one more aniline moiety removed compared to **C1**. Both **C1** and **C2** exhibited no appreciable antibiotic activity with their MIC's higher than 64 μg mL$^{-1}$ toward both ATCC43300 and ATCC19606 (Fig. 7a). Second, the peripheral group of the rhodamine core or the adamantine moiety could be modified without inhibiting the antibiotic activity as long as the rhodamine core remains intact, as showcased with **RD71-74** (Fig. 7b, c). The aforementioned hypothesis was further supported by the following experimental results that none of the starting materials (**S1-S69**) for preparations of these rhodamine analogs (**RD1-RD69**) display antibiotic activity (MIC > 64 μg mL$^{-1}$) (Supplementary Table 3). This reiterates the necessity of the rhodamine core for antibiotic activity. Additional support to this hypothesis comes from the ten compounds listed in Fig. 7d. These compounds predominantly adopt the ring-closed form and therefore they do not actually have the rhodamine motif. And indeed, none of them have been found to exhibit antibiotic activity during the phenotypic screening. The discovery of the above structure-activity relationship is important as it provides a general guideline for further structural modification, if warranted in the future drug-development endeavors.

**Discussion**

Diversity-oriented synthesis of xanthene library attracted attention for discovery of original bioactive compounds through phenotypic screening. This has promoted search for a mild xanthene synthesis allowing facile installation of functional, topographical and stereochemical diversity. We have devised a mild, one-step, high-yielding and readily scalable xanthene synthesis, by reacting a rationally constructed dilithium species

**Table 3 Mutations present in RD53-resistant *A. baumannii***

| Mutation[a] | Gene name[b] | Definition | Amino acid change |
|---|---|---|---|
| SNPs | | | |
| G →C | *kup* | KUP system potassium uptake protein | A373P |
| A →T | *mexB* | multidrug efflux pump | S617T |
| A →G | *mexB* | multidrug efflux pump | L149P |
| G →T | *trkH, trkG, ktrB* | trk system potassium uptake protein | S14R |
| G →A | *npdA* | NAD-dependent deacetylase | A14T |
| C →T | *rsmJ* | 16S rRNA (guanine1516-N2)-methyltransferase | A228T |
| C →T | *hisD* | histidinol dehydrogenase | V257A |
| G →A | *etk-wzc* | tyrosine-protein kinase Etk/Wzc | D569N |
| InDel | | | |
| GAACCT→- | DPO3G | DNA polymerase III subunit gamma/tau | 441_442del |

[a]Nucleotide change (genetic location in *A. baumannii* strain ATCC19606)
[b]Locus name as annotated in *A. baumannii* strain ATCC19606

with alkyl/aryl/alkenyl substituted esters, anhydrides, lactones and even some lactams and synthesized a focused library of xanthene dyes with 70 members. Through in vitro screening against various multi-drug-resistant pathogens, we found that it is a rich source of hit compounds exhibiting potent bactericidal activity. Particularly, **RD22** and **RD53** are two antibacterial hits exhibiting favorable properties for further drug development. First, the growth of a wide-spectrum of Gram-positive and Gram-negative pathogens was inhibited by **RD22** and **RD53** with a low MIC. They are a potent bactericidal agent and a low dose at (2.5–10)× MIC could reduce the concentration of methicillin-resistant *S. aureus*, vancomycin-resistant E. *faecalis*, polymyxin E resistant *A. baumannii* by 4 orders of magnitude within ca. 4 h, more quickly than the current first-line and last-resort antibiotics, e.g. vancomycin, linezolid, tigecycline, and daptomycin. While the pathogens readily acquire resistance toward these existing antibiotics, exposing the cells toward sub-MIC concentration of **RD22** and **RD53** did not readily induce emergence of resistance. Fourth, they have a large therapeutic index of over 100, which is defined by the Lysis20/MIC. Our work is reminiscent of the discovery of Prontosil, the first antibiotic, from an azo dye library by Gerhard Domagk. The successful treatment of streptococcus infection of her daughter foreran what was later known as the Golden Age of Antibiotics. With this manuscript, we want to share with the community that dye libraries of high structural diversity could potentially be that sought-after chemical space of alternative antibiotics when no other natural or unnatural chemical space are currently offering abundant hit compounds for further drug development.

## Methods

**Construction and characterization of the rhodamine library.** Details of synthesis and characterization of **RD1–RD74, C1–C2** employed in this manuscript were included in the Supplementary Methods section (Supplementary Figure 383-393). Structures of substrates (**S1–S69**) can be found in Supplementary Figure 1. General procedures B was illustrated as the general synthetic route for the preparation of rhodamine **RD1–72** via nucleophilic condensation of a dilithium reagent **1** with various substrates **S1–S70**. The UV-Vis absorption and fluorescence emission spectra in pH = 7.4 PBS with 1% DMSO was shown in Supplementary Figures 2-69, and spectral properties were summarized in Supplementary Table 1. For $^1$H-/$^{13}$C-NMR and MS spectra see Supplementary Figures 135-382.

**Bacterial cell culturing.** The following bacterial strains were used: *Staphylococcus aureus* ATCC25923 (American type culture collection, Beijing Zhongyuan Ltd., cat. no. 25923), CMCC26003 (National Center for Medical Culture Collections, cat. no. 26003), ATCC43300 (American type culture collection, Beijing Zhongyuan Ltd., cat. no. 43300), *Enterococcus faecalis* ATCC29212 (American type culture collection, Beijing Zhongyuan Ltd., cat. no. 29212), ATCC51299 (American type culture collection, Beijing Zhongyuan Ltd., cat. no. 51299), *Enterococcus faecium* ATCC35667 (American type culture collection, Beijing Zhongyuan Ltd., cat. no. 35667), *Staphylococcus epidermidis* CMCC26069 (National Center for Medical

Culture Collections, cat. no. 26069), *Streptococcus pyogenes* CMCC32006 (National Center for Medical Culture Collections, cat. no. 32006), *Helicobacter pylori* Sydney Strain 1 (BioVector NTCC Inc., cat. no. HpSS1), *Acinetobacter baumannii* ATCC19606 (American type culture collection, Beijing Zhongyuan Ltd., cat. no. 19606), *Pseudomonas aeruginosa* ATCC27853 (American type culture collection, Beijing Zhongyuan Ltd., cat. no. 27853), *Escherichia coli* ATCC25922 (American type culture collection, Beijing Zhongyuan Ltd., cat. no. 25922), *Shigella flexneri* ATCC51081 (American type culture collection, Beijing Zhongyuan Ltd., cat. no. 51081). The clinical strain isolated from patients was provided and tested by Shanghai Huashan Hospital. Bacteria were cultured in nutrient media, either cation-adjusted Mueller-Hinton broth (*S.aureus, E.faecalis, E.faecium, S.epidermidis, S.pyogenes, A.baumannii, P.aeruginosa, E.coli, S.flexneri*) at 37 °C in shaking flasks or Brucella agar supplemented with 5% defibrinated sheep blood (*H.pylori*) under microaerophile conditions at 37 °C. The abbreviations and strains of all the organisms used in this article are summarized in Supplementary Table 4.

**Minimum inhibitory concentration (MIC).** MIC was determined by broth microdilution for most species and by agar dilution method for *Helicobacter pylori* according to CLSI guidelines. Cell concentration was adjusted to approximately $5 \times 10^5$ cells per mL. For broth microdilution, the test medium was cation-adjusted Mueller-Hinton broth (MHB). After 20 h of incubation at 37 °C, the MIC was defined as the lowest concentration of antibiotic with no visible growth. As for agar dilution method, the test medium was supplemented Brucella agar plates. The plates were incubated under microaerophile conditions for 48 h at 37 °C. The MIC was defined as the lowest concentration that completely inhibited visible growth as compared to the drug-free control. Expanded antibacterial spectrum to clinical isolates was tested at Shanghai Huashan Hospital. Experiments were performed with biological replicates. MIC's of reference antibiotics against different organisms are summarized in Supplementary Table 5.

**Time-kill curves.** An overnight culture of cells (methicillin resistant *S. aureus* ATCC43300; vancomycin resistant *Enterococcus* ATCC51299; polymyxin E-resistant *A. baumannii* which was induced from ATCC19606) was diluted 1:10,000 in MHB and incubated at 37 °C with aeration at 220 r.p.m. for 2 h (early exponential). Bacteria were then challenged with **RD22** or **RD53** at 2.5 × MIC, 5 × MIC, 10 × MIC (a desirable concentration at the site of infection) comparing with antibiotics clinically used in culture tubes at 37 °C and 220 r.p.m. At intervals, 100 μl aliquots were removed, tenfold serially diluted suspensions were plated on MHA plates and incubated at 37 °C overnight. *H. pylori* SS1 cultured overnight (c. $2 \times 10^8$ c.f.u. per mL) of Brucella broth supplemented with 2% heat-inactivated fetal bovine serum (FBS broth) was diluted 1:100 with the same medium containing 1 × MIC, 2 × MIC, 4 × MIC, 8 × MIC **RD22** or **RD53** comparing with antibiotics clinically used at 37 °C, under microaerophile conditions and 220 r.p.m. At intervals, 100 μL aliquots were removed, tenfold serially diluted suspensions were plated on supplemented Brucella agar plates and incubated at 37 °C for 48 h. Colonies were counted and c.f.u. per mL was calculated. Experiments were performed with biological replicates.

**Resistance study.** An overnight culture of cells (methicillin resistant *S. aureus* ATCC43300; vancomycin resistant *Enterococcus* ATCC51299; *A. baumannii* ATCC19606 were grown in the presence of subinhibitory levels (0.25 × MIC) of **RD22** or **RD53** in order to increase the probability of obtaining resistant mutants. Antibiotics clinically used were included as a control. Next, the cells were added to **RD22** or **RD53** present at 0.25 × MIC, 0.5 × MIC, 1 × MIC, 2 × MIC and 4 × MIC. At 24 h intervals, the cultures were checked for growth. Cultures from the highest concentrations ($C_n$, n = passage number) that allowed growth (OD$_{600}$ ≥ 2) were diluted 1:100 into fresh media containing 0.25 × $C_n$, 0.5 × $C_n$, 1 × $C_n$, 2 × $C_n$ and

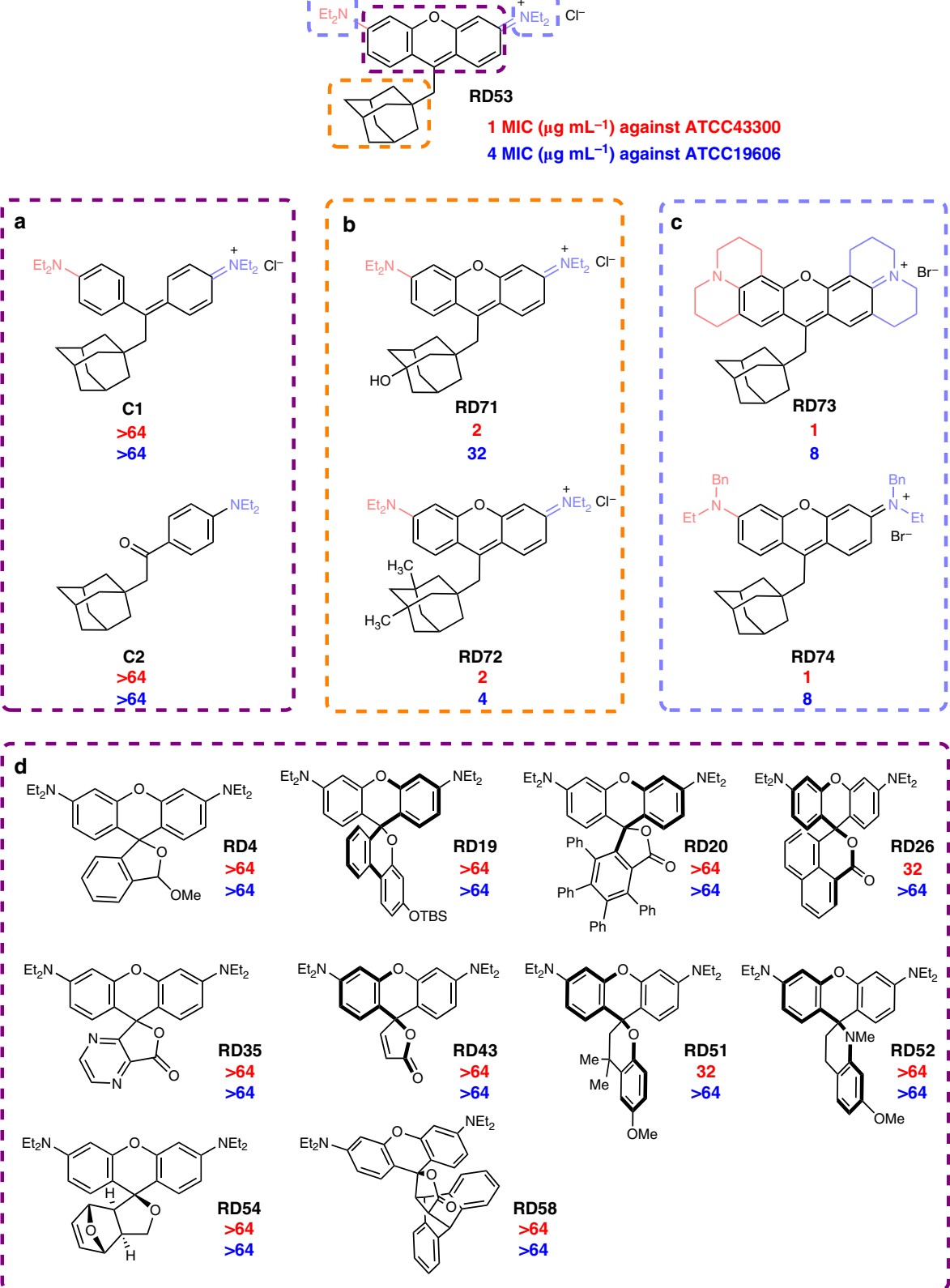

**Fig. 7** Structure-activity relationship investigation of the rhodamine library. **a** RD53 analogs without the rhodamine core; **b** RD53 analogs with the adamantine moiety modified; **c** RD53 analogs with the diethylamino group modified. **d** Rhodamine dyes mainly adopted the ring-close forms exhibit no antibacterial activity against ATCC43300 or ATCC19606, with MIC > 64 µg mL$^{-1}$

$4 \times C_n$ of **RD22** or **RD53**. This serial passaging was repeated daily for 20 days. Experiments were performed with biological replicates. Results of absolute fold change of MIC of 30 colonies purified from the 5th and 10th passage were shown in Supplementary Table 7. Results of spontaneous frequency of resistance study were shown in Supplementary Table 8.

**Scanning electron microscope (SEM) analysis for MRSA and *A. baumannii*.** The cells were grown in the MHB broth medium to around $10^7$ c.f.u. per mL and collected before drug treatment and 4 h after treatment of $8 \times$ MIC **RD53** (8 µg mL$^{-1}$ for MRSA and 32 µg mL$^{-1}$ for *A. baumannii*). The cells were fixed in 2.5% glutaraldehyde (prepared in PBS buffer) for 2 h, and washed with PBS buffer twice after fixation. Then the cells were dehydrated in a series of ethanol solutions which are 30%, 40%, 50%, 60%, 70%, 80%, 90%, 100% ethanol, sequentially for 15 min each, and finally suspended in pure tert-butanol for 15 min. The cells were critical-point dried and coated with gold following by scanning electron microscope analysis (FEI Nova NanoSEM 230). The SEM was operated in the high vacuum at an accelerating voltage of 7.5 kV.

**Haemolytic activity: red blood cell (RBC) lysis assay (Lysis20).** Lysis20 values of **RD22** and **RD53** were measured as the concentration of compound that lyses 20% or less of sheep red blood cells (RBC). The therapeutic index (Lysis20/MIC) was over 100 when comparing the MIC against MRSA to the observed Lysis20. RBC lysis assays were performed on mechanically defibrinated sheep blood. 1.5 mL of blood was placed into a microcentrifuge tube and centrifuged at $8609 \times g$ for 10 min. The supernatant was removed and the cells were resuspended with 1 mL of 5% glucose solution. The suspension was centrifuged as previously, the supernatant was removed, and cells were resuspended two more times. The final cell suspension was diluted twentyfold with 5% glucose solution. The twentyfold suspension dilution was then aliquoted into microcentrifuge tubes containing compound serially diluted in 5% glucose solution. Triton X (1% by volume) served as a positive control (100% lysis marker) and sterile 5% glucose solution served as a negative control (0% lysis marker). Samples were then placed in an incubator at 37 °C and shaken at 200 r.p.m. After 1 h, the samples were centrifuged at $8609 \times g$ for 10 min. The absorbance of the supernatant was measured with a UV spectrometer at a 330 nm wavelength.

**Mammalian cytotoxicity of RD53.** Cytotoxicity was determined on normal human kidney 2 cells HK-2 cells human renal proximal tubular cells (Cell Bank of Chinese Academy of Science, cat. no. SCSP-511), HUVEC Primary Umbilical Vein Endothelial Cells (American type culture collection, cat. no. PCS-100-013) and HeLa human cervical carcinoma cells (Cell Bank of Chinese Academy of Science, cat. no. TCHu 187). Briefly, HK-2 and HeLa cells were cultured in RPMI-1640 medium supplemented with FBS, and HUVEC were cultured in DMEM, high glucose medium supplemented with FBS. When cells reached ~80% confluence, they were harvested and resuspended in the growth medium to ~$2.0 \times 10^5$ cells mL$^{-1}$. 100 µL cells were inoculated into 96 well plate and incubated at 37 °C with 5% $CO_2$ for 4.5 h. Then, 0.25 µL of twofold serial compound dilutions in DMSO to 99.75 µL of media were added to the plate and incubated at 37 °C with 5% $CO_2$ for 43 h. For cell viability test, 20 µL 0.5% MTT solution were added to the plate and the supernatant was discarded after 4 h incubation. Then 150 µL DMSO were added to dissolve formazan and the A490nm (OD490) was measured. 50% cell cytotoxicity ($CC_{50}$) was analyzed in GraphPad Prism. Results are shown in Supplementary Figure 134.

**Whole-genome sequencing.** A single colony of various bacteria was grown until the $OD_{600}$ value reaches 0.3. The Genomic DNA was extracted and subjected to paired-ends Illumina sequencing. The whole genome sequencing of **RD53**-Ab-20th and ATCC19606 was acquired on an Illumina HiSeq 2000 system (Illumina Inc; San Diego, USA). The manufacturer's protocol (NEBNext® Ultra™ DNA Library Prep Kit for Illumina®) was followed to construct the Next Generation Sequencing Library Preparations. The extracted genomic DNA (1 µg) was sonicated to random fragments containing 500 bp or less. The End Prep Enzyme Mix was employed to treat the resulting fragments for end-repairing, 5′-phosphorylation and dA-tailing in one step. T-A ligation was carried out to add adaptors to both ends. Adaptor-ligated DNA was then sized-selected with AxyPrep Mag PCR Clean-up to recover the fragments of ca. 410 bp, the approximate insert size of which is 350 bp. The resulting samples were subsequently amplified by PCR for 8 cycles. The PCR products were treated with the AxyPrep Mag PCR Clean-up kit, validated with Agilent 2100 Bioanalyzer, and quantified by Qubit2.0 Fluorometer. Then libraries with different indices were multiplexed and loaded onto an Illumina HiSeq instrument. Sequencing was performed using a 2x150 paired-end (PE) configuration. The image analysis and base calling were carried out with the HiSeq Control Software (HCS) + OLB + GAPipeline-1.6 (Illumina) on the HiSeq instrument. The sequences of adaptors, PCR primers, content of N bases more than 10% and bases with a quality lower than 20 were removed using Cutadapt (V1.9.1). The clean data was mapped to the reference genome with BWA (V0.7.12) and duplicate-removed by Picard (V1.119). The Unified Genotyper calls single-nucleotide variant (SNV), insertions/deletions (InDel) with the GATK (V3.4.6) software. Annotation for

SNV/InDel was performed with Annovar (V21 Feb 2013). Pindel and CNVnator were employed to analyze genomic structure variation.

## Data availability

The data that support the findings of this study are available from the corresponding authors on request.

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

## Acknowledgements

This work is supported by the Fundamental Research Funds for the Central Universities (No. WY1516017) and the National Natural Science Foundation of China (Nos. 21572061, 21822805, and 81573329).

## Author contributions

Y.Y., D.C. and X.Q. conceived the projects. X.L. designed the synthetic method, synthesis **RD1**-**72** and acquired the UV-Vis absorption and fluorescence properties. Y.X., L.G., Z.L. and T.H. consynthesized **RD73**-**74** and **C1**-**C2**. L.Q. measured the MIC's of standardized bacteria, collected the dose-kill curves and carried out the induced resistance test and did the SEM imaging of bacteria. L.S., X.D., J.B. and J.W. contributed to phenotypic screenings. X.L., Y.T., Y.Z., J.Z. and W.Z. did the confocal imaging of mammalian cells. X.W. and H.L. calculated the diversity index. F.H. measured the MIC's of the clinical strains. All authors contributed to manuscript preparations.

## Additional information

**Competing interests:** All authors declare no competing interests.

