## [Peer Review File · Nature Communications]

Reviewers' comments:

Reviewer #1 (Remarks to the Author):

In their manuscript, Chen, Qian, Yang and co-workers describe the diversity-oriented synthesis of a library of rhodamines by a nucleophilic condensation of a dilithium reagent mainly with esters and anhydrides. The fluorescence and the bactericidal properties of this library was investigated in detail and intriguingly, one of the compounds (RD53) was particularly active and a potential lead for a novel class of antibiotics. The synthetic method is a mild alternative to the traditional thermal reactions, allowing even late-stage transformations of natural products and pharmaceuticals.

The introduction clearly describes the motivation for preparing novel antibiotics, but it only provides a short outline of the specific concepts of the present work. Instead, it refers to the recent publications from the groups of Lavis and Sparr. While the diversity of the obtained products and their properties are very interesting, some issues about the synthetic protocol and the yields have to be addressed:

(1)

As most of the rhodamines are salts, the anion is required to calculate the yield. However, as noted by Sparr and Lavis, a standard work-up gives different salts or salt mixtures. These anions would therefore have to be either quantitatively determined, exchanged with a suitable anion exchange resin, or avoided by using the acid in the work-up that corresponds the precursor halogens.

(2)

Because of the diversity of the library, the physical properties of the products vary tremendously: the isolation and handling of the individual products should therefore be described in detail.

Overall, this work contributes both to the field of diversity-oriented synthesis and to antibiotics research. The impact of this results would be however much improved by a specific description of the principal concepts. To sufficiently describe the experimental results, detailed protocols for the individual product should be provided. Furthermore, for determining the yields, the salts require an unambiguous anion assignment.

As the introduction and the unequivocal synthesis of the salts require a major amount of work, this reviewer recommends a later re-submission of this manuscript.

Reviewer #2 (Remarks to the Author):

The manuscript by Luo et al. describes the synthesis of a relatively small but focused library of rhodamine dyes and highlight that some of the analogs, and one in particular, possess some antibacterial potency against a variety of bacterial species. The versatility of the synthetic chemistry was highlighted with a range of 'substrates' that enabled the installation of significant substituent diversity at C-9 and extends the work of another laboratory. This reviewer feels that in a broader discipline journal such as Nature Communications, the synthetic aspects could be condensed and some of the information included in the Supplemental Information.

- The diversity analysis presented claims differentiation from the previous library but it would be useful to briefly include information, possibly scores from totally unrelated libraries, to enable the non-specialist reader to understand the relative significance of the 18.875 to 3.228 difference calculated between the 2 rhodamine libraries.
- The finding of some antibacterial activity with a subset of the analogs provides the best rationale for publication of this manuscript in a cross-discipline journal. The initial cutoff of 8 mg/L and 16 mg/L for MRSA and *A. baumannii*, respectively is a modest cutoff for potential starting points to identify lead molecules. The authors chose to further evaluate RD53 based on claims of best activity, but should explain their rationale for not exploring RD22 which exhibited the same initial profile as RD53. Having 2 structurally different exemplars from their library would have strengthened their claim that this chemistry platform could be useful for the identification of novel leads.
- The *A. baumannii* staining in Figure 7 (panel C) is at a resolution that is difficult to confirm the authors conclusions. After zooming in on the figure it appears that some of the cells in Figure 7C have intracellular staining whereas other at the top of the frame do not. Given the difference in shape between the species, and the thicker cell membrane of MRSA, the claim on lines 223-224 about the location of the target needs to be substantiated further. Either quantitative measurements of a statistically relevant number of individual cells, use of mutant strains, or even hybridization of non-denatured immobilized proteins after separation of cellular fractions with RD53 should be explored.
- The activity of RD53 against penicillin non-susceptible *Streptococcus pneumoniae* should be tested and reported – these are on both the WHO and CDC pathogen lists.
- The activity of RD53 against eukaryotic cells in a cytotoxicity assay should be provided to ensure that the antibacterial activity is not positively correlated with general cytotoxicity (and also erythrocyte assays to confirm absence of membrane disruption).
- Are the graphs in Figure 9 (and supplementary figure S132) simply the concentration of RD53 that supports growth upon passage? Individual colonies should be purified from each passage and characterized for absolute fold change with respect to RD53 (and control drugs). Multiple mutational lineages can co-exist and propagate in these types of population resistance experiments and the cultures should be purified to accurately understand independent mutations.
- The resistant mutants that emerged upon serial passage with *A. baumannii* and *E. coli* need to be characterized (Figure 9C and S132A). What is their phenotypic profile with respect to susceptibility to other antibacterial drugs? Furthermore, the

genome of these mutants should be sequenced to identify the source of the mutation that reduces the susceptibility as this may also provide significant insight to the identity of the target.

- Given the speed of resistance attainment in *A. baumannii*, spontaneous frequency of resistance experiments should be performed in this species, as well as the others, to better characterize the resistance risk.
- The authors claim that that RD53 does not acquire resistance in MRSA and use vancomycin as a surrogate (line 274-275). This is not an appropriate comparison as the experimental conditions used do not enable the horizontal transfer of the van cluster that provides this resistance.
- Line 294 claims that RD53 has favorable properties for drug development. To make this claim the authors need to provide data on some other characteristics that would be important in this regard, such as metabolic and chemical stability, plasma protein binding and solubility.
- Lines 301-307 – although the examples provided on antibacterial drugs coming from small focused libraries is tantalizing, there are just as many that require significant chemistry programs to optimize. The fact is true that RD53 could progress and coming from a small library is not an exclusionary factor, this claim would be significantly more compelling if additional data on drug-like properties are generated and presented as suggested.

Minor grammatical suggestions:

Throughout – the tense should be standardized appropriately to be consistent.

Line 32-33 – reword for clarity

Line 35-36 – Change to something like “The genes encoding resistance to the natural products that are present within the original organisms can then be horizontally transferred to pathogenic microbes enabling resistance to emerge”

Line 36 – adjuvants should be better defined.

Line 56/60 – reference 45 is Ahn et al., Grimm et al, and Fischer & Sparr

Line 230 – “drug-resistant bacteria classified by...”

Line 238/240 – spelling of bacterial names

Reviewer #3 (Remarks to the Author):

The submitted manuscript reports a Diversity-Oriented Rhodamine Library that applies to find new motif for bactericidal agents. Although there are interesting data especially for the diverse rhodamine library construction, the flow of the work does not support the logic of rhodamine as the key component for bactericidal activity. The main body of the work is construction of rhodamine library, whose diversity exceeds the previously reported library. This is an excellent achievement and very solid with spectral and computational analysis. After that, it is not clear why RD53 was picked from the library and examined for detailed bactericidal activity. There were some other similar level primary hits, such as RD22. The authors just claimed that RD53 was the best compound ‘because of its substituent, i.e. adamantyl moiety.’ Yes, it is true that adamantyl moiety has been known to have antibacterial effect. Then, what is the role of rhodamine part? Isn’t it reasonable to make adamantyl library rather than rhodamine library? Other than the bactericidal effect, the authors did not suggest other applicable possibilities of the library. Therefore, an additional experiment either for adamantyl library or other application of rhodamine library to keep the current claim & title of the paper.

Minor comments:

1. If authors want to show novel bactericidal activity from the library member, the mechanism of the hit compounds should be added. If the activity is from the known motif, the comparison data with known drug should be included. For example, most of antibiotics have bacterial specific target (cell wall synthesis, folic acid metabolism and so on). However, there is no information about the mechanism of RD53 in antibiotic activity. Since the direction of this paper is not well-organized, authors should specify the purpose and importance of library, and then reorganize the experiments.
2. The authors mentioned the inhibitory effect of RD53 is correlated with cytosolic compartments of bacteria through 3 min staining data (Figure 7). The time scale is not suitable to observe the inhibitory effect. Authors should show the localization of RD53 depending on time until the cytotoxic point and confirm it with bacteria live/death marker.
3. In Table 1, most of MICs show too wide range against pathogen bacteria. It would be better to repeat the experiment and confine the values into smaller range of better defined number.
4. The authors showed that RD53 has a bactericidal activity comparing to three other standard antibiotics by colony counting and claimed that RD53 is suitable for clinical application. It is a little too premature claim and at least the sensitivity of RD53 on mammalian cell should be compared to bacteria.
5. There is too many information scattered in many figures and table apart from their explanation, so it is difficult to follow the contents. It would be helpful to rearrange them in more consistent manner, for example, by putting the description right in front of or behind figures.
6. The authors mentioned about low-yielding compounds. It would be better to improve the yield and include the compound into the library, or just leave it out.
7. It would be good to show the full name of abbreviation at the first time of the usage. For example, MIC was used without explanation when it first came out.
8. It would be better to change the word ‘overmine’, which is not common word and it is difficult to catch the meaning.
9. In figure 9, it would be better to number each figure for better understanding.

Reviewers' comments:

Reviewer #1 (Remarks to the Author):

- (1) As most of the rhodamines are salts, the anion is required to calculate the yield. However, as noted by Sparr and Lavis, a standard work-up gives different salts or salt mixtures. These anions would therefore have to be either quantitatively determined, exchanged with a suitable anion exchange resin, or avoided by using the acid in the work-up that corresponds the precursor halogens.

Reply:

In our original synthetic procedure, we dried the organic layer after extraction with $\text{MgSO}_4 \cdot \text{H}_2\text{O}$. So, it is indeed possible that both chloride and sulfate may be present as the counterion.

We have revised purification procedure, which we resynthesized those rhodamine analogs insalt form. To be specific, we used CaCl_2 to dry the organic layer after extraction to avoid the introduction of sulfate anion. This way, we ensured that rhodamine dyes are paired with Cl^- only. We have accordingly recalculated the yields.

- (2) Because of the diversity of the library, the physical properties of the products vary tremendously: the isolation and handling of the individual products should therefore be described in detail.

Reply:

We have updated the synthetic procedures of each compound in the SI, following the suggestion of the reviewer 1.

All the rhodamine analogs were prepared with the same protocol, i.e. stirring 100 mg of various substrates (**S1-S70**) with 1.3 equiv. of dilithium reagent (1) at -78°C in anhydrous THF for 2 hrs. Each reaction was worked up with the same protocol, i.e. quenching with saturated NH_4Cl solution, extraction with CH_2Cl_2 , drying with CaCl_2 , evaporation under reduced pressure, and column chromatographic purification. We note that different eluent was used for different rhodamine dyes. Therefore, the composition of eluent was specified individually.

- (3) The impact of these results would be however much improved by a specific description of the principal concepts.

Reply:

The primary challenge for the development of novel antibiotic drugs is the lack of an alternative molecular space capable of providing abundant hits exhibiting favorable antibiotic activities for further optimization after the soil actinomycetes has been over-exploited.

It is the objective of this manuscript to share with the community that *the diversity-oriented rhodamine library is potentially such a sought-after molecular space for discovery of antibiotic hit compounds*. We have reorganized the entire manuscript to support the viability of this principle concept.

First, with the chemistry section, we report the facile construction of a focused rhodamine library and quantitatively analyzed the structural diversity of the library with a well-established chemometric protocol.

Second, with the phenotypic screening section, we confirm that (1) the existence of surprisingly abundant hits exhibits antibiotic activity, (2) few shortlisted hits (**RD22** and **RD53**) exhibit high inhibitory activity and broad spectrum toward 11/12 WHO prioritized drug-resistant pathogens, (3) **RD22** and **RD53** has potent bactericidal activities against Gram-positive/negative pathogens, and (4) **RD22** and **RD53** do not readily induce bacterial resistance.

Third, the potential mechanism of action of **RD53** was probed with SEM, cross-resistance experiments, and whole-genome sequencing of the **RD53**-resistant *A. baumannii*.

Fourth, the structure-activity relationship was investigated to further substantiate that the rhodamine core is the key pharmacologically active motif to render antibiotic activity.

We feel that the principle concept of this manuscript is presented clearly and convincingly with such an organization of the manuscript.

- (4) To sufficiently describe the experimental results, detailed protocols for the individual product should be provided. Furthermore, for determining the yields, the salts require an unambiguous anion assignment.

Reply:

Among the 70 compounds (**RD1-RD70**) of this library, 10 of them, i.e. **RD4**, **RD19**, **RD20**, **RD26**, **RD35**, **RD43**, **RD51**, **RD52**, **RD54**, **RD58**, were found to predominantly exist in the ring-closed form by ¹H-NMR, And 7 of them were in their inner salt forms, i.e. **RD6**, **RD18**, **RD28**, **RD50**, **RD55**, **RD57**, **RD59**. The synthetic procedures and yields of these 17 compound do not need revision.

As for the other 43 compounds, we attempted to resynthesize them all, following a revised workup and purification procedure to avoid the presence of other anions but Cl⁻. To be specific, CaCl₂ was used in the new syntheses, instead of MgSO₄, as the dehydrating agent of the organic layer after extraction to avoid potential introduction of SO₄²⁻ as the counterion of the rhodamine dye.

Very interestingly, essentially identical amount of pure dyes were obtained with the first four dyes we resynthesized (**RD1**, **RD2**, **RD3**, and **RD7**), compared to the previous synthesis with the original workup and purification protocol. This makes us wonder if the chloride is the counterion of the rhodamine dyes, with either CaCl₂ or MgSO₄ as the dehydrating agent. We then tried another three (**RD25**, **RD49**, and **RD60**) to verify our hypothesis. Indeed, the same amount of dyes (within a very reasonable range) was obtained.

We updated the SI with the revised procedure.

We acknowledge that our original yields were calculated with molecular weight without Cl⁻. Therefore, the original yields were indeed questionable. We have recalculated these yields with the correct molecular weight. We thank the reviewer for the help with this issue.

Reviewer #2 (Remarks to the Author):

- (1) This reviewer feels that in a broader discipline journal such as Nature Communications, the synthetic aspects could be condensed and some of the information included in the Supplemental Information.

Reply:

We have greatly reduced the discussion of the synthesis by more than half, following the suggestion of the reviewer.

- (2) The diversity analysis presented claims differentiation from the previous library but it would be useful to briefly include information, possibly scores from totally unrelated libraries, to enable the non-specialist reader to understand the relative significance of the 18.875 to 3.228 difference calculated between the 2 rhodamine libraries.

Reply:

The Extended-connectivity fingerprints (ECFPs) method is a commonly employed fingerprint topological methodology to capture precise atom environment substructural features. The ECFP feature count for a chemical library is widely used as a rapid assessment of its structural diversity and can be used to compare the structural diversity in different libraries, especially in literatures related to phenotypic screening and drug development. One of the original papers reporting this method has been cited over 1170 times (Reference 39: Rogers, D. & Hahn, M. Extended-connectivity fingerprints. *J. Chem. Inf. Model.* **50**, 742-754 (2010)).

The ECFPs method can be used to different level of sophistication. In our study, ECFP_6 was employed, where 6 means the effective diameter of the largest feature. The total number of features is the sum of the features discovered at 6 iteration steps from all the members of a library. Then the diversity number fingerprint features (DNFPF) was defined as the total number of fingerprint features divided by the number of molecules.

$$DNFPF = \frac{\text{the total number of fingerprint features}}{\text{the number of molecules}}$$

In reference 39, four representative existing chemical libraries were assessed with the ECFP_6 method. The first library contains 50,000 compounds, all of which were randomly selected from the Derwent World Drug Index. This library is a chemical library of high structural diversity and expected to yield a high DNFPF index. This library was calculated to contain over 750,000 ECFP features. Therefore, a DNFPF value of ca. 25 was found.

In the same manuscript, another library of 50,000 compounds was assessed. All the compounds of this library were selected from a combinatorial library with an indole core. The structural diversity of this library is low and a small DNFPF value is expected. Indeed, a DNFPF value less than 1.4 was found.

We hope that these two literature examples can help the audience to make sense the relative significance of 18.875 vs 3.228.

- (3) The finding of some antibacterial activity with a subset of the analogs provides the best rationale for publication of this manuscript in a cross-discipline journal. The initial cutoff of 8 mg/L and 16 mg/L for MRSA and *A. baumannii*, respectively is a modest cutoff for potential starting points to identify lead molecules. The authors chose to further evaluate **RD53** based on claims of best activity, but should explain their rationale for not exploring **RD22** which exhibited the same initial profile as **RD53**. Having 2 structurally different examples from their library would have strengthened their claim that this chemistry platform could be useful for the identification of novel leads.

Reply:

After we finished preparation of this rhodamine library, we actually performed three rounds of phenotypic screening.

Round 1. All analogs (**RD1-RD70**) were screened against MRSA and *A. baumannii* to check their MIC's, which are tabulated in Figure 4. Ones with the lowest MIC's, hence highest inhibitory activity, against either MRSA or *A. baumannii* were shortlisted.

Round 2. The best few from the round 1 were screened against an additional list of pathogens, i.e. three Gram-positive and three Gram-negative pathogens, to have a preliminary touch of their antibiotic spectrum. **RD22** and **RD53** are the two best candidates based on their antibiotic potency and spectrum width.

Round 3. Originally, only **RD53** but not **RD22** was chosen for further wide-spectrum testing, because it does exhibit slightly overall better performance and because we wish to reduce the efforts/cost of biological evaluations.

However, the reviewer 2 is absolutely correct that having more active analogs will greatly strengthen our claim that this chemical space is potentially a viable source of antibiotic compounds. Therefore, we further tested **RD22** for its spectrum-width of antibiotic activity (Table 2), time-dependent killing curves against MRSA, VRE and polymyxin E resistant *A. baumannii* (Figure 5), and resistance acquisition of pathogens during serial passaging (Figure 6). All the data suggest that **RD22** is as promising as **RD53** in all aspects of antibacterial properties, if not better.

- (4) The *A. baumannii* staining in Figure 7 (panel C) is at a resolution that is difficult to confirm the authors conclusions. After zooming in on the figure it appears that some of the cells in Figure 7C have intracellular staining whereas other at the top of the frame do not. Given the difference in shape between the species, and the thicker cell membrane of MRSA, the claim on lines 223-224 about the location of the target needs to be substantiated further. Either quantitative measurements of a statistically relevant number of individual cells, use of mutant strains, or even hybridization of non-denatured immobilized proteins after separation of cellular fractions with **RD53** should be explored.

Reply:

In our original manuscript, the fluorescence images of **RD53** were included to offer some insights on the potential localization of target. However, we agree with the reviewer that the tentative conclusion that “*the inhibitory effect of RD53 toward the bacteria likely involves the interaction of RD53 with cytosolic components of bacteria*” needs further substantiation.

It is not routine practice to tentatively elucidate the potential antibacterial target via fluorescence imaging. However, these fluorescence images are viable data and could be of potential significance, therefore we simply moved the fluorescence imaging results to SI without further discussion.

After we received the comments and revision suggestions, we have carried out four biological studies to probe the potential mechanism of action of **RD53**, i.e. 1) *haemolysis of red blood cells*, 2) *SEM imaging of RD53 treated MRSA and A. baumannii*, 3) *the whole genome sequencing of RD53 resistant A. baumannii*, and 4) *cross-resistance of RD53 and five antibiotics against A. baumannii*.

The aforementioned data are not yet sufficient to nail down the underlying mechanism of action of **RD53**. However, it is sure that the destruction of cell wall or membrane is not responsible. Interaction of **RD53** with cytosolic components of bacteria does.

- (5) The activity of **RD53** against penicillin non-susceptible *Streptococcus pneumoniae* should be tested and reported; these are on both the WHO and CDC pathogen lists.

Reply:

The three clinical strains of *S. pneumoniae* are penicillin-resistant strains, i.e. PRSP. We forgot to include this information in the original submitted manuscript.

RD53 has a MIC of 2-4 µg/mL toward PRSP, and **RD22** an even lower MIC of 0.5-1 µg/mL. Therefore, both **RD22** and **RD53** exhibit high inhibitory potency toward PRSP.

- (6) The activity of **RD53** against eukaryotic cells in a cytotoxicity assay should be provided to ensure that the antibacterial activity is not positively correlated with general cytotoxicity (and also erythrocyte assays to confirm absence of membrane disruption).

Reply:

We assayed the haemolytic activity of both **RD53** and **RD22** against human red blood cells (RBCs). A Lysis₂₀ value of above 100 µg/mL was measured for both **RD53** and **RD22**. Therefore a high therapeutic index, i.e. Lysis₂₀/MIC of above 100 was calculated. Note: Lysis₂₀ values are the concentration of a compound that lyses 20% or less of red blood cells (RBC).

The CC₅₀ value of **RD53** was measured against human umbilical vein endothelial cells (HUVECs), to be 3.912 µg/mL. We admit that this value is not yet optimal if **RD53** is intended for systemic use. However, there's ample room for further structural modification to reduce its cytotoxicity during the in-depth lead optimization process. Second, **RD53** itself could readily find practical use against dermal infections, in which case its potential cytotoxicity is less a concern. Therefore, we believe that **RD53** along with other members of this rhodamine library is a major progress in the field of antibiotic development and still deserves the prestige of *Nat. Comm.*

- (7) Are the graphs in Figure 9 (and supplementary figure S132) simply the concentration of **RD53** that supports growth upon passage? Individual colonies should be purified from each passage and characterized for absolute fold change with respect to **RD53** (and control drugs). Multiple mutational lineages can co-exist and propagate in these types of population resistance experiments and the cultures should be purified to accurately understand independent mutations.

Reply:

The concentration of **RD53** in original Figure 9 (which is figure 4 in the revised manuscript) means the highest concentration (tested up to 128-256xMIC) that bacteria could grow to a minimum OD₆₀₀ of 0.2 at 24 hour intervals. We agree that **multiple mutational lineages can co-exist and propagate in these types of population resistance experiments**. So we purified the 5th and 10th passage of heritable resistant bacteria such as *Acinetobacter baumannii* and *Escherichia coli* on MHA plates and randomly selected thirty individual colonies to test the MICs of control drugs, **RD22** and **RD53** to characterize for absolute fold change with respect to those compounds. Before measuring MIC, each individual colony was inoculated to antibiotic-free broth for three times. We found that the different colonies of bacteria population gave different MIC's. The MIC's of control drugs to almost all individual colonies of heritable resistant bacteria were higher than C_x in Figure 4. In comparison, more colonies of **RD22** and **RD53** resistant bacteria could not grow in C_x.

In conclusion, this experiment indicates that these bacteria had lower tendency of heritable resistance acquisition toward **RD22** and **RD53** than what Figure 4 has shown.

strain	antibiotic	passage	Colonies growable in C _x /tested colonies
A. baumannii	RD22	5 th	29/30
		10 th	4/30
	RD53	5 th	23/30
		10 th	3/30
	PmE	5 th	30/30
		10 th	30/30
	CIP	5 th	30/30
		10 th	30/30
TGC	5 th	30/30	
	10 th	30/30	
E. coli	RD53	5 th	30/30
		10 th	5/30
	CRO	5 th	30/30
		10 th	30/30
TCY	5 th	25/30	

CIP	10 th	29/30
	5 th	30/30
	10 th	30/30

- (8) The resistant mutants that emerged upon serial passage with *A. baumannii* and *E. coli* need to be characterized (Figure 9C and S132A). What is their phenotypic profile with respect to susceptibility to other antibacterial drugs? Furthermore, the genome of these mutants should be sequenced to identify the source of the mutation that reduces the susceptibility as this may also provide significant insight to the identity of the target.

Reply:

We have checked the phenotypic profile of **RD53**-resistant *A. baumannii* against five other clinical antibiotics, e.g. tigecycline (TGC), ciprofloxacin (CIP), polymyxin E (PmE), levofloxacin (LEV), ampicillin-sulbactam (SAM). As we can see from the following table, the MIC of *A. baumannii* enhanced by 32 fold from 4 mg/mL to 128 mg/mL. But, the MIC of this **RD53**-resistant strain remained essentially unchanged toward five other antibiotics. This suggests that **RD53** likely works via a different mechanism of action than these five antibiotics.

Strain	ATCC19606	RD53 -Ab-20th
RD53	4	128
TGC	1	0.5
CIP	1	0.5
PmE	1	1
LEV	1	0.25
SAM	8	16

We further carried out the whole-genome sequencing of this **RD53**-resistant *A. baumannii*. Comparative analysis revealed the existence of eight nonsynonymous single nucleotide variants (SNV) and one nonframeshift deletion, which belong to resistance-associated genes encoding efflux pumps, metabolism DNA replication or transcriptional regulation related proteins. The mutation in *rsmJ* encoding 16S rRNA (guanine1516-N2)-methyltransferase is particularly interesting. Notably, the mutation in 16S rRNA methyltransferase has been shown to cause kasugamycin resistance in *E. coli*.

Since selective pressure on antibacterial targets could also be assessed by genomic profile of mutational resistant bacteria. So, the genomic analysis detailed in the following table offers valuable clues on the potential mechanism of action of **RD53**.

Mutation ^a	Gene Name ^b	Definition	Amino Acid Change

SNPs	G → C	kup	KUP system potassium uptake protein	A373P
	A → T	mexB	multidrug efflux pump	S617T
	A → G	mexB	multidrug efflux pump	L149P
	G → T	trkH, trkG, ktrB	trk system potassium uptake protein	S14R
	G → A	npdA	NAD-dependent deacetylase	A14T
	C → T	rsmJ	16S rRNA (guanine1516-N2)-methyltransferase	A228T
	C → T	hisD	histidinol dehydrogenase	V257A
	G → A	etk-wzc	tyrosine-protein kinase Etk/Wzc	D569N
InDel	GAACCT→-	DPO3G	DNA polymerase III subunit gamma/tau	441_442del

- (9) Given the speed of resistance attainment in *A. baumannii*, spontaneous frequency of resistance experiments should be performed in this species, as well as the others, to better characterize the resistance risk.

Reply:

We have tested the spontaneous frequency in *A. baumannii*, MRSA, *E. coli* and *Enterococcus* to better characterize the resistance risk. The spontaneous frequency in *A. baumannii*, MRSA, *E. coli* and *Enterococcus* S. aureus was determined by plating 10^9 cells per plate onto MHA plates containing 2xMIC, 4xMIC, and 8xMIC **RD53**, **RD22** or control drugs. After 48 hours of incubation, the plates were examined for colonies. Spontaneous frequency = the number of colonies grown on the agar plates containing the compound/ 10^9 . As we can see from the following table, there were no colonies of *Enterococcus* on plates with $\geq 2^*$ MIC. Note that the intrinsic variability in determining MIC is also 2 fold. So, even at a very low concentration of the compound, there was no resistance development. While the spontaneous frequencies of MRSA to **RD22** and **RD53** were lower than levofloxacin, the spontaneous frequencies of *A. baumannii* to **RD22** and **RD53** were lower than polymyxinE (PmE) and the spontaneous frequencies of *E. coli* to **RD22** and **RD53** were lower than ciprofloxacin (CIP), ceftriaxone (CRO) and tetracycline (TCY). So, the conclusion of this experiment is that **RD22** and **RD53** were less prone to induce resistance than some control drugs.

Strain	antibiotic	MIC(μ g/ml)	spontaneous frequency of resistance		
			2*MIC	4*MIC	8*MIC
ATCC43300	RD53	0.5	10^{-7}	10^{-8}	$< 10^{-9}$
	RD22	0.25	10^{-6}	10^{-7}	10^{-8}
	VAN	2	$5 \cdot 10^{-8}$	10^{-8}	10^{-9}

	LEV	0.25	10 ⁻⁵	10 ⁻⁶	10 ⁻⁷
ATCC19606	RD53	8	2*10 ⁻⁶	< 10 ⁻⁹	< 10 ⁻⁹
	RD22	8	3*10 ⁻⁵	1*10 ⁻⁵	1*10 ⁻⁶
	PmE	1	10 ⁻⁴	2*10 ⁻⁵	10 ⁻⁶
	TGC	2	10 ⁻⁸	< 10 ⁻⁹	< 10 ⁻⁹
	CIP	1	10 ⁻⁵	2*10 ⁻⁶	< 10 ⁻⁹
ATCC51299	RD53	2	< 10 ⁻⁹	< 10 ⁻⁹	< 10 ⁻⁹
	RD22	2	< 10 ⁻⁹	< 10 ⁻⁹	< 10 ⁻⁹
	TGC	0.125	< 10 ⁻⁹	< 10 ⁻⁹	< 10 ⁻⁹
	CIP	0.5	< 10 ⁻⁹	< 10 ⁻⁹	< 10 ⁻⁹
	LNZ	2	< 10 ⁻⁹	< 10 ⁻⁹	< 10 ⁻⁹
ATCC29212	RD53	1	< 10 ⁻⁹	< 10 ⁻⁹	< 10 ⁻⁹
	RD22	1	< 10 ⁻⁹	< 10 ⁻⁹	< 10 ⁻⁹
	TGC	0.125	< 10 ⁻⁹	< 10 ⁻⁹	< 10 ⁻⁹
	CIP	1	< 10 ⁻⁹	< 10 ⁻⁹	< 10 ⁻⁹
	LNZ	2	< 10 ⁻⁹	< 10 ⁻⁹	< 10 ⁻⁹
ATCC25922	RD53	8	10 ⁻⁸	10 ⁻⁹	< 10 ⁻⁹
	RD22	32	< 10 ⁻⁹	< 10 ⁻⁹	< 10 ⁻⁹
	CIP	0.015625	10 ⁻⁷	< 10 ⁻⁹	< 10 ⁻⁹
	CRO	0.0625	10 ⁻⁵	10 ⁻⁶	10 ⁻⁷
	TCY	2	10 ⁻⁶	10 ⁻⁷	10 ⁻⁸

- (10) The authors claim that that **RD53** does not acquire resistance in MRSA and use vancomycin as a surrogate (line274-275). This is not an appropriate comparison as the experimental conditions used do not enable the horizontal transfer of the van cluster that provides this resistance.

Reply:

We agreed that acquired resistance mainly contains two mechanisms, which is caused by importing foreign genetic elements and mutations in genome. So we correct acquire resistance to mutational resistance which is more accurate.

- (11) Line 294 claims that **RD53** has favorable properties for drug development. To make this claim the authors need to provide data on some other characteristics that would be important in this regard, such as metabolic and chemical stability, plasma protein binding and solubility. Lines 301-307; although the examples provided on antibacterial drugs coming from small focused libraries is tantalizing, there are just as many that require significant chemistry programs to optimize. The fact is true that **RD53** could progress and coming from a small library is not an exclusionary factor, this claim would be significantly more compelling if additional data on drug-like properties are generated and presented as suggested.

Reply:

We thank the reviewer for this comment.

These two properties are favorable for drug development.

1. The hemolytic activity of **RD53** and **RD22** is low with a Lysis20 values above 100 µg/mL.
2. Both **RD22** and **RD53** have a sufficient water solubility of above 5 mg/mL.

We also checked the cytotoxicity of **RD53** toward mammalian cells. For example, a CC₅₀ of 3 µg/mL was recorded for human umbilical vein endothelial cells (HUVECs). We admit that this value is not yet optimal if **RD53** is intended for systemic use. However, there's ample room for further structural modification to reduce its cytotoxicity during the in-depth lead

optimization process. Second, **RD53** itself could readily find practical use against dermal infections, in which case its potential cytotoxicity is less a concern.

We checked the chemical stability of **RD53** and **RD22**. Both are stable toward nucleophiles (e.g. H₂O, H₂S, GSH, Cys), oxidants (e.g. H₂O₂), and reductants (e.g. NADPH, GSH) up to their physiological levels, since the absorbance of their solution remain unchanged upon addition of these chemicals. These data are trivial and not included in the SI.

However, the metabolic stability of **RD22** and **RD53** were not studied since we feel this experiment should be performed only after we have confirmed the *in vivo* antibiotic activity in near future.

We were not able to provide all the requested data yet at this moment. However, we do believe that, with all the data we have accumulated, **RD53** could be regarded as a feasible lead compound potentially suitable for further chemical optimization.

Therefore, we believe that **RD53** along with other members of this rhodamine library is a major progress in the field of antibiotic development and still deserves the prestige of *Nat. Comm.*

- (12) Throughout; the tense should be standardized appropriately to be consistent.

Reply:

We have standardized the tense of the manuscript.

- (13) Line 32-33; reword for clarity

Reply:

The sentence has been rewritten into “*The resistant strains may readily mutate to resist these newer analogs if their existing resistance mechanisms do not already exhibit partial cross-effectiveness.*”

- (14) Line 35-36; Change to something like: “The genes encoding resistance to the natural products that are present within the original organisms can then be horizontally transferred to pathogenic microbes enabling resistance to emerge”;

Reply:

We deeply thank the reviewer for help with prosing of our manuscript.

- (15) Line 36; adjuvants should be better defined.

Reply:

The sentence regarding “adjuvants” is not very relevant to the current context and therefore removed from the text.

- (16) Line 56/60; reference 45 is Ahn et al., Grimm et al, and Fischer & Sparr

Reply:

We thank the reviewer for correcting us with this error with referencing.

- (17) Line 230; “drug-resistant bacteria classified by...”;

Reply:

The original writing has been corrected.

- (18) Line 238/240; spelling of bacterial names

Reply:

The spelling of “flexner” is now “flexneri” and “methicilin” now “methicillin”.

Reviewer #3 (Remarks to the Author):

- (1) Although there are interesting data especially for the diverse rhodamine library construction, the flow of the work does not support the logic of rhodamine as the key component for bactericidal activity. Since the direction of this paper is not well-organized, authors should specify the purpose and importance of library, and then reorganize the experiments.

Reply:

We agree with the reviewer that the manuscript was not well organized and the original submission does not seem to have a clear take-home message for the audience. We have carefully substantiated the manuscript with more judiciously designed experiments and carefully reorganized the manuscript. We believe that the current version of the manuscript is much improved and thank the reviewer for their inputs.

The outline of the manuscript is summarized in the following diagram for a quick evaluation of the flow/logic of the manuscript for the convenience of the three reviewers.

Manuscript Outline		
Introduction		
Results	Chemistry	Synthesis of the rhodamine library
		Diversity evaluation
	Phenotypic screening	Inhibitory potency and spectrum
		Bactericidal activity
		Resistance study
	Probing Mechanism of action	
Structure-Activity relationship		
Discussion		

- (2) The authors just claimed that **RD53** was the best compound because of its substituent, i.e. adamantyl moiety. Yes, it is true that adamantyl moiety has been known to have antibacterial effect. Then, what is the role of rhodamine part? Isn't it reasonable to make adamantyl library rather than rhodamine library? Therefore, an additional experiment either for adamantyl library or other application of rhodamine library to keep the current claim & title of the paper.

Reply:

Thank the reviewer to point this out.

It is the rhodamine that delivers the antibacterial activity. We have carried a thorough structure-activity relationship to verify this.

First, the precursors for preparations of rhodamine dyes did not exhibit any antibiotic activity, regardless the presence of adamantyl moiety.

Second, those compounds adopted the ring-closed form exhibited no appreciable antibiotic activity with their MIC's higher than 64 $\mu\text{g/mL}$ toward both ATCC43300 and ATCC19606.

- (3) It is not clear why **RD53** was picked from the library and examined for detailed bactericidal activity. There were some other similar level primary hits, such as **RD22**.

Reply:

Originally, only **RD53** but not **RD22** was chosen for further wide-spectrum testing, because it does exhibit slightly overall better performance. And, that time we wish to reduce the efforts/cost of biological evaluations.

However, the reviewer 2 and the reviewer 3 are absolutely correct that having more active analogs will greatly strengthen our claim that this chemical space is potentially a viable source of antibiotic compounds. Therefore, we further tested **RD22** for its spectrum-width of antibiotic activity (Table 2), time-dependent killing curves against MRSA, VRE and polymyxin E resistant *A. baumannii* (Figure 5), and resistance study of pathogens during serial passaging (Figure 6). All the data suggest that **RD22** is as promising as **RD53** in all aspects of antibacterial properties, if not better.

- (4) If authors want to show novel bactericidal activity from the library member, the mechanism of the hit compounds should be added. If the activity is from the known motif, the comparison data with known drug should be included. For example, most of antibiotics have bacterial specific target (cell wall synthesis, folic acid metabolism and so on). However, there is no information about the mechanism of **RD53** in antibiotic activity.

Reply:

After we received the comments and revision suggestions, we have carried out four biological studies to probe the potential mechanism of action of **RD53**, i.e. 1) haemolysis of red blood cells, 2) SEM imaging of **RD53** treated MRSA and *A. baumannii*, 3) the whole genome

sequencing of **RD53** resistant *A. baumannii*, and 4) cross-resistance of **RD53** and five antibiotics against *A. baumannii*.

The aforementioned data are not yet sufficient to nail down the underlying mechanism of action of **RD53**. However, it is sure that the destruction of cell wall or membrane is not responsible. Interaction of **RD53** with cytosolic components of bacteria does.

- (5) Other than the bactericidal effect, the authors did not suggest other applicable possibilities of the library.

Reply:

Potential applications of this dye library are diverse. Considering the structure diversity, the library is best use toward phenotypic screening for hit compounds against agriculturally or medically significant microbes. Second, the dyes in this library could be used as a fluorescent label or probes as well.

- (6) The authors mentioned the inhibitory effect of **RD53** is correlated with cytosolic compartments of bacteria through 3 min staining data (Figure 7). The time scale is not suitable to observe the inhibitory effect. Authors should show the localization of **RD53** depending on time until the cytotoxic point and confirm it with bacteria live/death marker.

Reply:

In our original manuscript, the fluorescence images of **RD53** were included to offer some insights on the potential localization of target. However, we agree with the reviewer that the tentative conclusion that “*the inhibitory effect of RD53 toward the bacteria likely involves the interaction of RD53 was cytosolic components of bacteria*” needs further substantiation.

It is not routine practice to tentatively elucidate the potential antibacterial target via fluorescence imaging. However, these fluorescence images are viable data and could be of potential significance, therefore we simply moved the fluorescence imaging results to SI without further discussion.

After we received the comments and revision suggestions, we have carried out four biological studies to probe the potential mechanism of action of **RD53**, i.e. 1) *hemolysis of red blood cells*, 2) *TEM imaging of RD53 treated MRSA and A. baumannii*, 3) *the whole genome sequencing of RD53 resistant A. baumannii*, and 4) *cross-resistance of RD53 and five antibiotics against A. baumannii*.

The aforementioned data are not yet sufficient to nail down the underlying mechanism of action of **RD53**. However, it is sure that the destruction of cell wall or membrane is not responsible. Interaction of **RD53** with cytosolic components of bacteria does.

- (7) In Table 1, most of MICs show too wide range against pathogen bacteria. It would be better to repeat the experiment and confine the values into smaller range of better defined number.

Reply:

MIC was determined by broth microdilution for most species and by agar dilution method for helicobacter pylori according to CLSI guidelines. The MIC's are reported to be 256, 128, 64, 32, 16, 8, 4, 2, 1, 0.5, 0.25, ... in unit of $\mu\text{g/mL}$.

- (8) The authors showed that **RD53** has a bactericidal activity comparing to three other standard antibiotics by colony counting and claimed that **RD53** is suitable for clinical application. It is a little too premature claim and at least the sensitivity of **RD53** on mammalian cell should be compared to bacteria.

Reply:

We assayed the haemolytic activity of both **RD53** and **RD22** against human red blood cells (RBCs). A Lysis20 value of above 100 $\mu\text{g/mL}$ was measured for both **RD53** and **RD22**. Therefore a high therapeutic index, i.e. Lysis20/MIC of above 100 was calculated. Note:

Lysis₂₀ values are the concentration of a compound that lyses 20% or less of red blood cells (RBC).

The CC₅₀ value of **RD53** was measured against human umbilical vein endothelial cells (HUVECs), to be 3.912 µg/mL. We admit that this value is not yet optimal if **RD53** is intended for systemic use. However, there's ample room for further structural modification to reduce its cytotoxicity during the in-depth lead optimization process. Second, **RD53** itself could readily find practical use against dermal infections, in which case its potential cytotoxicity is less a concern. Therefore, we believe that **RD53** along with other members of this rhodamine library is a major progress in the field of antibiotic development and still deserves the prestige of *Nat. Comm.*

- (9) There is too many information scattered in many figures and table apart from their explanation, so it is difficult to follow the contents. It would be helpful to rearrange them in more consistent manner, for example, by putting the description right in front of or behind figures.

Reply:

We have thoroughly checked for this problem and fixed the relevant errors throughout the manuscript.

- (10) The authors mentioned about low-yielding compounds. It would be better to improve the yield and include the compound into the library, or just leave it out.

Reply:

This section is now removed following the suggestion of the reviewer.

- (11) It would be good to show the full name of abbreviation at the first time of the usage. For example, MIC was used without explanation when it first came out.

Reply:

We have checked for this issue and made correction following the suggestion of the reviewer.

- (12) It would be better to change the word 'overmine', which is not common word and it is difficult to catch the meaning.

Reply:

We have changed this word.

- (13) In figure 9, it would be better to number each figure for better understanding.

Reply:

We have renumbered all the figures and changed the figures and texts accordingly.

REVIEWERS' COMMENTS:

Reviewer #1 (Remarks to the Author):

The authors present a revised manuscript addressing all of the main recommendations of the reviewers. The experimental data was improved and the text is tightened up. In conclusion, this work now contributes greatly to the field of diversity oriented synthesis of rhodamines and related compounds. Furthermore, the discovery of a potent new bactericidal agent confirmed the diversification approach for chemical library synthesis. Therefore, acceptance for publication of this manuscript is recommended together with further language corrections.

Reviewer #2 (Remarks to the Author):

The authors have appropriately addressed this reviewer's questions and provided additional data and clarity where requested. If the few instances where additional data were not added, rational and appropriate justifications were provided and do not impact the suitability of the manuscript for publication.

Reviewer #3 (Remarks to the Author):

The manuscript has been improved by correcting and adding more experimental data. While some of the suggested tasks were not fully carried, e.g. mechanism elucidation or superior selectivity of antibacterial vs. normal host cells, if the current data is the best it can reach, I believe this revised manuscript is acceptable to Nat Comm.